

# Probing proton structure at the Large Hadron electron Collider

**Rabah Abdul Khalek[1], Shaun Bailey[2], Jun Gao[3], Lucian Harland-Lang[2⋆] and Juan Rojo[1]**

**1** Department of Physics and Astronomy, Vrije Universiteit, 1081 HV Amsterdam,
and Nikhef Theory Group, Science Park 105, 1098 XG Amsterdam, The Netherlands
**2** Rudolf Peierls Centre for Theoretical Physics, University of Oxford,
Clarendon Laboratory, Parks Road, Oxford OX1 3PU, United Kingdom
**3** Institute of Nuclear and Particle Physics, Shanghai Key Laboratory for Particle Physics and
Cosmology, School of Physics and Astronomy, Shanghai Jiao Tong University, Shanghai, and
Center for High Energy Physics, Peking University, Beijing 100871, China

⋆ lucian.harland-lang@physics.ox.ac.uk

## Abstract

For the foreseeable future, the exploration of the high-energy frontier will be the domain of the Large Hadron Collider (LHC). Of particular significance will be its high-luminosity upgrade (HL-LHC), which will operate until the mid-2030s. In this endeavour, for the full exploitation of the HL-LHC physics potential an improved understanding of the parton distribution functions (PDFs) of the proton is critical. The HL-LHC program would be uniquely complemented by the proposed Large Hadron electron Collider (LHeC), a high-energy lepton-proton and lepton-nucleus collider based at CERN. In this work, we build on our recent PDF projections for the HL-LHC to assess the constraining power of the LHeC measurements of inclusive and heavy quark structure functions. We find that the impact of the LHeC would be significant, reducing PDF uncertainties by up to an order of magnitude in comparison to state-of-the-art global fits. In comparison to the HL–LHC projections, the PDF constraints from the inclusive and semi–inclusive LHeC data are in general more significant for small and intermediate values of the momentum fraction $x$. At higher values of $x$, the impact of the LHeC and HL–LHC data is expected to be of a comparable size, with the HL–LHC constraints being more competitive in some cases, and the LHeC ones in others.


# 1   Introduction

The particle physics community is busy preparing for the extensive precision and discovery physics programme that will come from Run III of the LHC, and most significantly, for the major upgrade beginning in the mid-2020s, the High-Luminosity LHC (HL-LHC). Here, protons will be collided with an instantaneous luminosity a factor of five greater than the LHC and will accumulate up to ten times more data, resulting in an integrated luminosity of around $\mathcal{L} = 3$ ab$^{-1}$ for both the ATLAS and CMS detectors, and 300 fb$^{-1}$ for LHCb. The rich physics prospects of the HL-LHC, which will operate until at least 2035, have recently been analyzed in detail [1–5], considering physics within and beyond the Standard Model (SM), Higgs, flavour, and heavy ion physics.

In this context, a precise determination of the quark and gluon structure of the proton, as encoded in the parton distribution functions (PDFs) [6–8], is an essential ingredient for the success of the HL–LHC. Conversely, the HL–LHC itself offers an unprecedented opportunity to improve our understanding of proton structure. We recently analyzed in detail the HL-LHC potential to constrain the PDFs [9] by using projected measurements for a range of SM processes, from Drell-Yan to top quark pair and jet production. We found that PDF uncertainties on LHC processes can be reduced by a factor between two and five, depending on the specific flavour combination and on the assumptions about the experimental systematic uncertainties. Our PDF projections have already been used in a number of related HL–LHC studies, as reported in [1, 2].

A quite distinct possibility to improve our understanding of proton structure is the proposal to collide high energy electron and positron beams with the hadron beams from the HL–LHC. This facility, known as the Large Hadron Electron Collider (LHeC) [10, 11, 17], would run concurrently with the HL–LHC and be based on a new purpose–built detector at the designed interaction point. A key outcome of the LHeC operations would be a significantly larger and higher–energy dataset of lepton–proton collisions in comparison to the existing HERA structure function measurements [12]. Indeed, the latter to this day form the backbone of all PDF determinations [12–16], and thus the LHeC would provide the opportunity to greatly extend the precision and reach of HERA data in both $x$ and $Q^2$, highlighting its potential for PDF constraints. Moreover, these measurements would be taken in the relatively clean environment of lepton–proton collisions, where the corresponding theoretical predictions are known to a very high level of precision. It should also be emphasized here that the LHeC has a broad and exciting physics program which goes well beyond studies of the proton structure, including topics such as the characterization of the Higgs sector or the study of cold nuclear matter in the small-$x$ region, where new QCD dynamical regimes such as saturation are expected to appear.

Quantitative PDF projection studies based on LHeC pseudo–data have been presented previously [10, 17–19], where a sizeable reduction in the resultant PDF uncertainties is reported.

These LHeC PDF projections are based upon the HERAPDF-like input PDF parametrization and flavour assumptions [12], with some additional freedom in the input parametrisation added in the most recent studies [20, 21]. These baseline fits include constraints from the HERA structure function measurements, with in some cases the addition of a limited subset of collider data. However, different results may be obtained if a more flexible parametrization or alternative flavour assumptions are used, as shown for example in the study of [22] carried out in the NNPDF framework. In addition, the interplay of these constraints from the LHeC with the expected sensitivity from the HL–LHC [9] has not yet been studied. Thus a natural question to ask is how the projected sensitivity of a state-of-the-art global PDF determination will improve with data from the LHeC, and how this will complement the information contained in the measurements in $pp$ collisions provided by the HL–LHC.

In this paper, we study in detail the projected sensitivity of the LHeC for constraining PDFs within the framework of a global analysis. We follow the strategy presented in [9], starting from the PDF4LHC15 [23–25] baseline set, and quantify the expected impact of the LHeC measurements both individually and combined with the information provided by the HL–LHC. For the LHeC pseudodata we use the most recent publicly available official LHeC projections on the expected statistical and systematic errors, and choice of binning [26, 27] (see also [28] for further details) As we will demonstrate, the expected constraints from the LHeC are significant and fully complementary with those from the HL–LHC. When included simultaneously, a significant reduction in PDF uncertainties in the entire relevant kinematical range for the momentum fraction $x$ is achieved, with beneficial implications for LHC phenomenology.

In addition, in order to understand some of the methodological systematic effects which may be at play when performing such a profiling study, we also assess the impact of adopting the more restrictive HERAPDF input parameterisation as our baseline PDF set. The slightly subtle issue here is that when one generates pseudodata assuming an underlying PDF parameterisation one is implicitly assuming that it will be possible to describe the actual data with this parameterisation. If this assumption is too strong, that is the underlying parameterisation is too restrictive, then one is liable to overestimate the impact of the data. Consistent with this, we find a more marked reduction of the PDF uncertainties for the rather restrictive HERAPDF baseline in comparison to the global PDF4LHC case. To clarify this further, we present pure 'LHeC–only' fits, that is modifying the Hessian prior to effectively remove the impact of the data in the baselines, and isolate the impact of the parameterisation alone. These are therefore directly comparable to other LHeC–only projections; we in particular take a tolerance of $T = 1$ here. We again find significantly smaller projected errors when assuming the HERAPDF parameterisation in comparison to the more flexible PDF4LHC one.

The outline of this paper is as follows. In Sect. 2 we present the main features of the projected LHeC pseudo–data and the theory settings that will be used for the QCD analysis. In Sect. 3 we review the Hessian profiling formalism which is used to quantify the constraints on the PDFs of the LHeC pseudo–data. In Sect. 4 we present the PDF projections for the LHeC, both individually and in combination with the HL–LHC. In Sect. 5 we study in detail the impact of varying the tolerance and the flexibility of the underlying PDF parameterisation on the projected constraints. Finally, in Sect 6 we conclude.

## 2  Pseudo–data generation and theory calculations

In this section we present the details of the LHeC pseudo–data that will be used for our PDF projections, as well as the settings required to evaluate the corresponding theory predictions for the LHeC inclusive and heavy quark structure functions.

## 2.1 LHeC pseudo–data

For the LHeC dataset, we use the most recent publicly available official LHeC projections [26] (see also [28] for further details) for electron and positron neutral-current (NC) and charged-current (CC) scattering. The main features of the pseudo–data sets we consider are summarised in Table 2.1, along with the corresponding integrated luminosities and kinematic reach. While the nominal high energy data ($E_p = 7$ TeV) provides the dominant PDF constraints, the lower energy ($E_p = 1$ TeV) data extends the acceptance to higher $x$ and provides a handle on the longitudinal structure function, $F_L$, and hence the gluon PDF (we note that further variations in the electron and/or proton energy will provide additional constraints on this, as well as on other novel low–$x$ QCD phenomena). The charm and bottom heavy quark NC structure function pseudo–data provide additional constraints on the gluon. In addition, charm production in $e^- p$ CC scattering provides important information on the anti-strange quark distributions via the $\bar{s} + W \to \bar{c}$ process. We do not include charm production in $e^+ p$ CC scattering, as the corresponding pseudo-datasets are not currently publicly available, though this would provide an additional constraint on the strange quark PDF. We apply a kinematic cut of $Q \geq 2$ GeV to ensure that the fitted data lie in the range where perturbative QCD calculations can be reliably applied. We note that in what follows we will sometimes refer to 'data' for brevity, but this is always understood to imply the pseudo-datasets as described above.

Table 2.1: Overview of the main features of the LHeC pseudo–data [26,27] included in our PDF projections. For each process, we indicate the kinematic coverage, the integrated luminosity, the proton energy, and the number of pseudo–data points, $N_{\text{dat}}$, after the $Q \geq 2$ GeV kinematic cut. Note that in all cases the incoming lepton energy is fixed to be $E_l = 60$ GeV. We ignore the effect of the incoming lepton beam polarization.

| Observable | $E_p$ | Kinematics | $N_{\text{dat}}$ | $\mathcal{L}_{\text{int}} \, [\text{ab}^{-1}]$ |
|---|---|---|---|---|
| $\tilde{\sigma}^{\text{NC}} \, (e^- p)$ | 7 TeV | $5 \times 10^{-6} \leq x \leq 0.8$, $5 \leq Q^2 \leq 10^6$ GeV$^2$ | 150 | 1.0 |
| $\tilde{\sigma}^{\text{CC}} \, (e^- p)$ | 7 TeV | $8.5 \times 10^{-5} \leq x \leq 0.8$, $10^2 \leq Q^2 \leq 10^6$ GeV$^2$ | 114 | 1.0 |
| $\tilde{\sigma}^{\text{NC}} \, (e^+ p)$ | 7 TeV | $5 \times 10^{-6} \leq x \leq 0.8$, $5 \leq Q^2 \leq 5 \times 10^5$ GeV$^2$ | 148 | 0.1 |
| $\tilde{\sigma}^{\text{CC}} \, (e^+ p)$ | 7 TeV | $8.5 \times 10^{-5} \leq x \leq 0.7$, $10^2 \leq Q^2 \leq 5 \times 10^5$ GeV$^2$ | 109 | 0.1 |
| $\tilde{\sigma}^{\text{NC}} \, (e^- p)$ | 1 TeV | $5 \times 10^{-5} \leq x \leq 0.8$, $2.2 \leq Q^2 \leq 10^5$ GeV$^2$ | 128 | 0.1 |
| $\tilde{\sigma}^{\text{CC}} \, (e^- p)$ | 1 TeV | $5 \times 10^{-4} \leq x \leq 0.8$, $10^2 \leq Q^2 \leq 10^5$ GeV$^2$ | 94 | 0.1 |
| $F_2^{c,\text{NC}} \, (e^- p)$ | 7 TeV | $7 \times 10^{-6} \leq x \leq 0.3$, $4 \leq Q^2 \leq 2 \times 10^5$ GeV$^2$ | 111 | 1.0 |
| $F_2^{b,\text{NC}} \, (e^- p)$ | 7 TeV | $3 \times 10^{-5} \leq x \leq 0.3$, $32 \leq Q^2 \leq 2 \times 10^5$ GeV$^2$ | 77 | 1.0 |
| $F_2^{c,\text{CC}} \, (e^- p)$ | 7 TeV | $10^{-4} \leq x \leq 0.25$, $10^2 \leq Q^2 \leq 10^5$ GeV$^2$ | 14 | 1.0 |
| Total | | | 945 | |

For the integrated luminosity that is expected to be collected by the LHeC, we take $\mathcal{L} = 1$ ab$^{-1}$ for the high energy $E_p = 7$ TeV electron and positron cross sections, corresponding to roughly a three orders of magnitude larger dataset in comparison to HERA. In fact, as most

of these measurements become quickly dominated by systematic errors, our results do not depend too sensitively on this specific choice. For the lower energy inclusive structure functions ($E_p = 1$ TeV) as well as for the semi-inclusive measurements we assume $\mathcal{L} = 0.1$ ab$^{-1}$. In total we have $N_{\text{dat}} = 945$ pseudo–data points that satisfy the $Q \geq 2$ GeV kinematic cut. For all of the LHeC pseudo–data, we consider the case of unpolarized lepton beams, neglecting the effect of lepton polarisation. While beam polarization is, for example, important for precision measurements of electroweak parameters such as the $W$ mass or the Weinberg angle $\sin \theta_W$, for PDF determination it is known that the impact of beam polarisation effects is small.

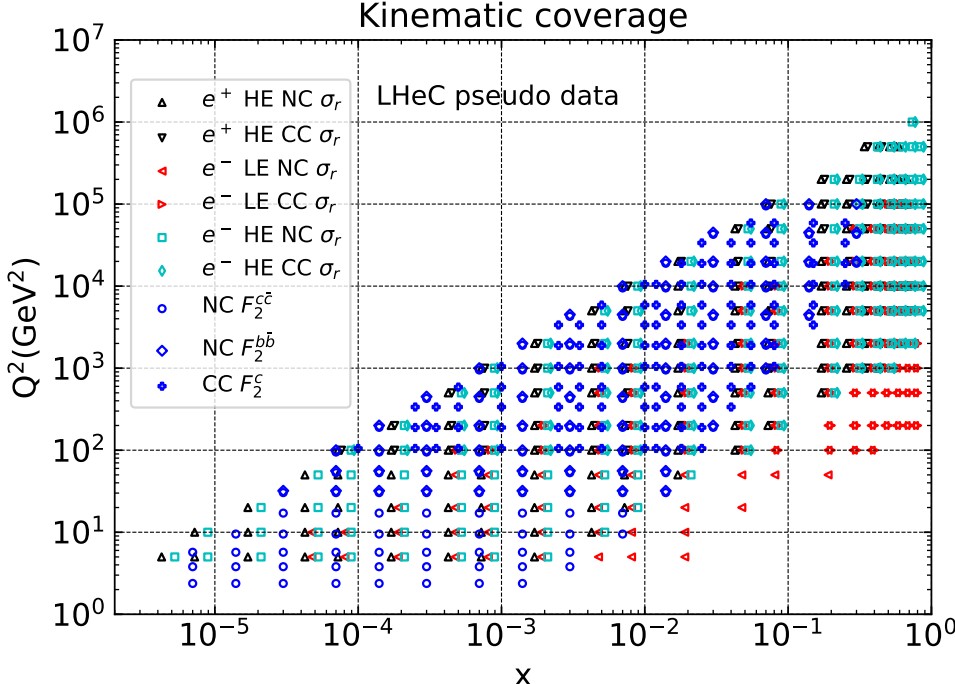

Figure 2.1: The kinematic coverage in the $(x, Q^2)$ plane of the LHeC pseudo–data [26, 27] included in the present analysis: the inclusive NC and CC structure functions both for high energy (HE) and low energy (LE) datasets, the NC charm and bottom semi-inclusive structure functions $F_2^{c\bar{c}}$ and $F_2^{b\bar{b}}$, and the CC charm structure functions $F_2^{c}$ providing direct information on the strange content of the proton.

The kinematic reach in the $(x, Q^2)$ plane of the LHeC pseudo–data is shown in Fig. 2.1. The reach in the perturbative region ($Q \geq 2$ GeV) is well below $x \approx 10^{-5}$ and extends up to $Q^2 \approx 10^6$ GeV$^2$ (that is, $Q \simeq 1$ TeV), increasing the HERA coverage by over an order of magnitude in both cases, via the factor $\sim 4$ increase in the collider centre-of-mass energy $\sqrt{s}$. Due to the heavy quark tagging requirements, the reach for semi-inclusive structure functions only extends up to $x \simeq 0.3$ in the large-$x$ region. Note that in addition to providing PDF information, the extended coverage of the LHeC in the high-$Q$ region would also provide novel opportunities for indirect searches for new physics beyond the Standard Model through precision measurements, see for example [10, 17, 29], as well as a rich program of Higgs production and decay studies.

The pseudo–data listed in Table 2.1 have been generated assuming a detector coverage with lepton rapidity $|\eta_l| \leq 5$ and inelasticity $0.001 \leq y \leq 0.95$. Systematic uncertainties due to the scattered electron (positron) energy scale and polar angle, hadronic energy scale,

calorimeter noise, radiative corrections, photoproduction background and a global efficiency error are included in a correlated way across the NC datasets, while a single global source of correlated systematic is taken across all CC datasets. In addition, an uncorrelated efficiency uncertainty of 0.5% is taken, while a fully correlated luminosity uncertainty of 1% is assumed. In the case of the semi-inclusive heavy-quark structure functions, there are two sources of systematics considered correlated across bins for both NC and CC production respectively.

We note that the statistical errors are generally an order of magnitude or more smaller than the systematic uncertainties, apart from close to kinematic boundaries, and hence as discussed above we would not expect our results to change significantly if somewhat smaller datasets are assumed. Indeed, we have explicitly verified the validity of this assumption by using instead an integrated luminosity of 0.3 ab$^{-1}$ for the case of high energy neutral-current electron scattering.

According to the above considerations, we then produce the pseudo–data values as usual by shifting the corresponding theory predictions by the appropriate experimental errors. In particular, the pseudo–data point $i$ is generated according to

$$\sigma_i^{\text{exp}} = \sigma_i^{\text{th}} \left( 1 + \delta_{\text{unc},i}^{\text{exp}} \cdot r_i + \sum_k \Gamma_{ik}^{\text{exp}} s_{k,i} \right), \qquad (2.1)$$

where $s_i$, $r_k$ are univariate Gaussian random numbers, $\Gamma_{ik}^{\text{exp}}$ is the $k$-th correlated systematic error and $\delta_{\text{unc},i}^{\text{exp}}$ is the total uncorrelated error for datapoint $i$. The $\sigma_i^{\text{th}}$ are the corresponding theoretical predictions computed using the baseline PDF set, which we discuss in more detail below. The $s_{k,i}$ random numbers are the same for all data points for which the $k$-th systematic error is fully correlated among them.

In addition to the processes listed in Table 2.1, there are additional PDF-sensitive measurements from the LHeC that one could consider in such an exercise. One example is jet production in electron-proton collisions, for which NNLO (and in some cases even N$^3$LO) QCD calculations are available [30–32], and that has also been studied at HERA [33], see also [34, 35]. While such jet production at the LHeC will provide additional information on the large-$x$ gluon, as the pseudo–data projections are not currently available, we do not consider these here. A further example is charm production in $e^+p$ CC scattering, which would provide a constraint on the strange quark PDF; we currently only include this process in $e^-p$ scattering, as the $e^+p$ pseudo-data projections are again not currently available. We note that these same choices have also been adopted by all other LHeC PDF projections carried out so far.

## 2.2 Theoretical calculations

For all the pseudo–data listed in Table 2.1 we have evaluated the corresponding theoretical predictions based on the PDF4LHC15 NNLO PDF set. Specifically, we use the Hessian version [23, 24] PDF4LHC15_100 composed of $N_{\text{eig}} = 100$ symmetric eigenvectors. This is of course not the only possible choice, and indeed as well as missing some of the more recent LHC and non–LHC data included in global fits, it omits the possibility for the LHeC to for example shed light on the different variable flavour schemes applied by the global fits included in PDF4LHC (see [36] for a general critical discussion). However, this remains a useful baseline, corresponding to the best available estimate for the current knowledge on proton structure from global PDF fits. Moreover, this is the same baseline PDF set used for our earlier HL–LHC projections [9], allowing us to directly compare the results of these with those based on the LHeC pseudo–data, as well as to combine the constraints provided by the two future facilities.

The LHeC structure functions have been evaluated at NLO using the APFEL program [37] with the FONLL-B general-mass variable-flavor-number scheme [38]. In a real PDF determi-

nation it would be important to include NNLO QCD corrections as well as in principle small-$x$ BFKL resummation corrections [22]. However in our case, where by construction the agreement between data and theory is good and we are only interested in evaluating the relative reduction in PDF uncertainties, this is not necessary. In particular, as the dominant PDF sensitivity is already contained within the NLO calculation, it is not necessary to include higher-order effects. On the other hand, theoretical uncertainties such as those arising from missing higher orders in the predictions that enter the PDF fit [39, 40], which are omitted here, may have some impact, given that as will be shown below the resultant PDF uncertainties are often at the per-mile level. In addition, for simplicity we do not consider the contribution from the choice of heavy quark masses or strong coupling on the PDF projections (see e.g. [15] for a study of these effects in a global PDF fit). In fact, as discussed most recently in [20], data from the LHeC can reduce the uncertainties on these inputs below the level where they will be expected to give a significant contribution to PDF uncertainties.

## 3   Hessian profiling and the role of tolerance

In this section we review the Hessian profiling formalism [41,42] which is used to estimate the constraints on the PDFs of the LHeC pseudo–data, following the general approach presented in [9]. After minimising with respect to the experimental nuisance parameters, the total $\chi^2$ due to $N_{\mathrm{dat}}$ pseudo–data points can be written as

$$
\begin{aligned}
\chi^2(\beta_{\mathrm{th}}) &= \sum_{i,j=1}^{N_{\mathrm{dat}}} \left( \sigma_i^{\mathrm{exp}} - \sigma_i^{\mathrm{th}} - \sum_k \sigma_i^{\mathrm{th}} \Gamma_{ik}^{\mathrm{th}} \beta_{k,\mathrm{th}} \right) C_{ij}^{-1} \left( \sigma_j^{\mathrm{exp}} - \sigma_j^{\mathrm{th}} - \sum_m \sigma_j^{\mathrm{th}} \Gamma_{jm}^{\mathrm{th}} \beta_{m,\mathrm{th}} \right) \\
&\quad + T^2 \sum_k \beta_{k,\mathrm{th}}^2 \,.
\end{aligned}
\tag{3.1}
$$

Here $\sigma_i^{\mathrm{exp}}$ and $\sigma_i^{\mathrm{th}}$ represent the central values of the pseudo–data and theory predictions, respectively. The $\beta_{k,\mathrm{th}}$ are the nuisance parameters corresponding to movement along the PDF eigenvectors, i.e. such that $\beta_{k,\mathrm{th}} = 0$ gives the prediction from the baseline PDF set, prior to profiling, while a non–zero value will result from the profiling itself. The matrix $\Gamma_{ik}^{\mathrm{th}}$ corresponds to the rate of change of the theory prediction $i$ with eigenvector $k$, encoding the effect of varying these nuisance parameters on the theory. The tolerance factor $T$ will be discussed further below.

The above expression for the $\chi^2$, Eq. (3.1), assumes that the final profiled theory prediction is sufficiently close to the original PDF prediction that we only have to expand to the first order in $\beta_{k,\mathrm{th}}$. That is, a linear approximation is taken, and higher order corrections beyond it are assumed to be negligible (see [43] for a discussion of such effects). As we are interested in a closure test, where this will by construction be true, profiling is expected to represent to very good approximation the result of performing an actual fit.

The experimental covariance matrix $C$ that enters the $\chi^2$ definition Eq. (3.1) is given by the following expression:

$$
C_{ij} = \delta_{ij} \left( \delta_{\mathrm{unc},i}^{\mathrm{exp}} \sigma_i^{\mathrm{th}} \right)^2 + \sum_k \sigma_i^{\mathrm{th}} \sigma_j^{\mathrm{th}} \Gamma_{ik}^{\mathrm{exp}} \Gamma_{jk}^{\mathrm{exp}} \,,
\tag{3.2}
$$

where $\Gamma_{ik}^{\mathrm{exp}}$ and $\delta_{\mathrm{unc},i}^{\mathrm{exp}}$ are defined as in Eq. (2.1). Note that as the impact of the uncorrelated errors are defined in terms of a fixed theoretical prediction (rather than of the fit output itself), our results are resilient with respect to the D'Agostini bias [44, 45].

Profiling then proceeds via the minimisation of Eq. (3.1) with respect to the Hessian PDF

nuisance parameters $\beta_{k,\text{th}}$. This can be performed analytically, resulting in

$$\beta_{k,\text{th}}^{\min} = -\sum_l H_{kl}^{-1} a_l \,, \tag{3.3}$$

where we have defined

$$H_{kl} = \sum_{i,j} \sigma_i^{\text{th}} \Gamma_{ik}^{\text{th}} C_{ij}^{-1} \sigma_j^{\text{th}} \Gamma_{jl}^{\text{th}} + T^2 \delta_{kl} \,, \tag{3.4}$$

$$a_k = \sum_{i,j} \sigma_i^{\text{th}} \Gamma_{ik}^{\text{th}} C_{ij}^{-1} \left( \sigma_j^{\text{th}} - \sigma_j^{\text{exp}} \right) \,. \tag{3.5}$$

The result of Eq. (3.3) represents a new shifted position in eigenvector space, and we can then readily construct a new set of profiled PDF parameters from this.

The matrix $H$ corresponds to the new Hessian matrix for the profiled fit, so that Eq. (3.1) can be rewritten as

$$\chi_{\text{profiled}}^2 = \chi_{\text{profiled}}^2\big|_{\beta_{k,\text{th}} = \beta_{k,\text{th}}^{\min}} + \sum_{kl} \delta\beta_{k,\text{th}} H_{kl} \delta\beta_{l,\text{th}} \,, \tag{3.6}$$

where $\delta\beta_{k,\text{th}} = \beta_{k,\text{th}} - \beta_{k,\text{th}}^{\min}$. This can be diagonalised in the usual way in terms of the eigenvectors $\vec{v}^{(k)}$ of $H$, with $k = 1, \cdots, N_{\text{eig}}$. This then provides a new set of PDF errors via the following expression

$$\delta\beta_{l,\text{th}}^{(k)} = T v_l^{(k)} \sqrt{\frac{1}{\hat{\epsilon}_k}} \,, \tag{3.7}$$

where the errors are treated as symmetric, and as above $v_l^{(k)}$ is the $l$th component of $k$th eigenvector of $H$, and $\hat{\epsilon}_k$ is the corresponding eigenvalue.

Now, in Eq. (3.7) we can see that the profiled PDF uncertainty carries an explicit dependence on the tolerance $T$, which ensures that the corresponding uncertainty as in (3.6) is determined via a $\Delta\chi^2 = T^2$ criterion. However, we can see from Eq. (3.4) that the new Hessian matrix also depends on the tolerance, and hence there is an additional implicit dependence on this in Eq. (3.7), via the eigenvalues $\hat{\epsilon}_k$. In particular, if we for example take the limit of a particularly unconstraining dataset, setting $C_{ij}^{-1} \sim 0$, then we have $\hat{\epsilon}_k \sim T^2$. This implies that the profiled PDF errors would be independent of $T$, being unchanged and equal to the original ones, as consistency would dictate. On the other hand, if we consider the case of a highly constraining dataset, where the first term in Eq. (3.4) dominates, the eigenvalues become independent of $T$ and the profiled PDF uncertainties scale linearly with it.

For the more realistic situation where the impact of a new dataset is of a similar order of magnitude to the PDF errors of the baseline set, i.e. the explicit constraints from the new dataset enters at a similar level to the implicit constraints from the datasets which result in the baseline set, the scaling with $T$ will be somewhere in between. Note that in this case the impact of the new dataset is also dictated by the size of the original tolerance as in Eq. (3.4); the larger the tolerance, the smaller the corresponding impact. This is entirely consistent with our physical expectation of the role the tolerance should play within a Hessian global PDF determination. As we will see below, the above considerations are relevant for the case of the LHeC dataset, where we will consider different values of the tolerance.

Unless otherwise stated, in the studies presented here, we use $T = 3$, which roughly corresponds to the average tolerance determined in the CT14 and MMHT14 analyses. This is the same choice as the one we adopted for our PDF projections with the HL–LHC pseudo–data. The resulting profiled PDF set[1] can be straightforwardly used for phenomenology using the

---

[1]Note that sometimes in this paper we will for brevity use the shorthand 'fit', but it always understood that a profiling has been performed rather than a full refit.

uncertainty prescription of symmetric Hessian sets, and the default output format is compliant with the LHAPDF interface [46].

# 4 Results

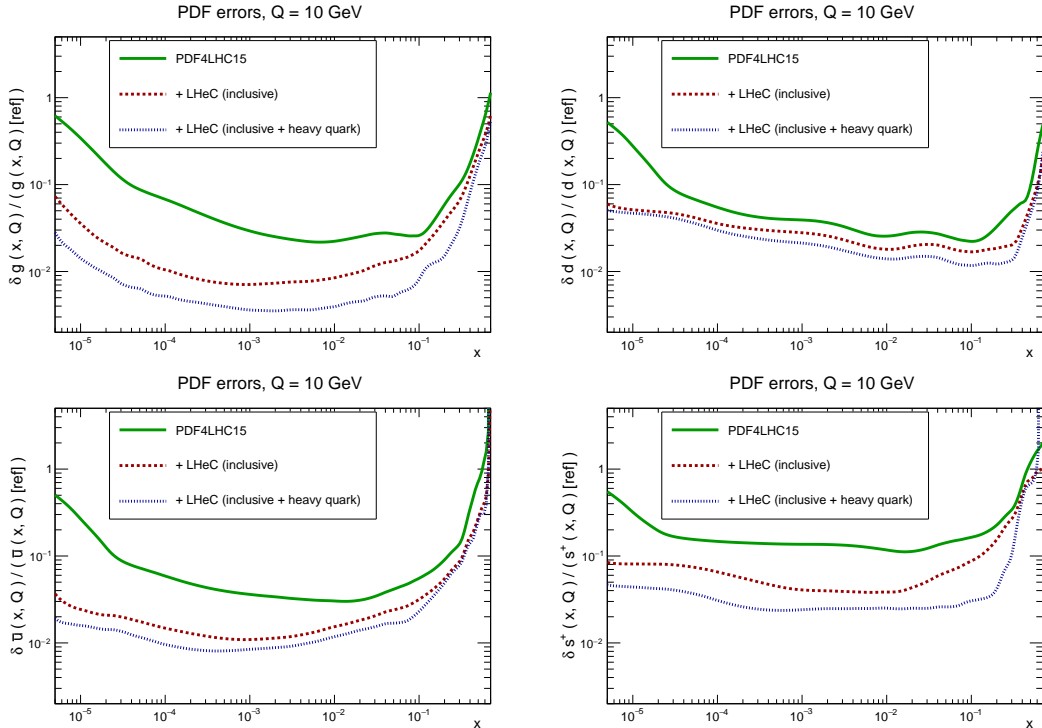

Figure 4.1: Impact of LHeC on the 1–$\sigma$ relative PDF uncertainties of the gluon, down quark, anti–up quark and strangeness distributions, with respect to the PDF4LHC15 baseline set. In this comparison, shown at $Q = 10$ GeV, we indicate the results of both profiling with the inclusive LHeC measurements alone and also with the semi-inclusive heavy quark structure functions.

We now present the main results of this work, namely quantifying the impact of the LHeC pseudo–data on the PDF4LHC15 baseline in different scenarios. We begin by considering the impact of the LHeC measurements alone with respect to the baseline, before combining these constraints with those provided by the HL–LHC. In Fig. 4.1 we indicate the impact of the LHeC structure function measurements, both for the inclusive dataset alone and in combination with the semi-inclusive heavy quark datasets, on the PDF uncertainties of the gluon, down quark, anti–up quark and strangeness distributions. We can see that the effect is in all cases significant, with the heavy quark data placing additional important constraints.

In the case of the gluon PDF, the inclusive pseudo–data already place significant constraints at low to intermediate $x$, through the precise direct measurements of their $Q^2$ slope, $\partial F_2 / \partial \ln Q^2$, as well as indirectly through the constraints on the quarks. The heavy quark data further reduce the uncertainties on the gluon, with the impact at high $x$ from the charm and beauty structure functions being most significant here. The inclusive data have some impact on the down quark, in particular at high $x$, where the uncertainties on the dominantly valence distribution are significantly reduced. A qualitatively similar, and somewhat larger, effect is found for the up quark, which we do not show here for brevity. The impact on the anti–up

from the inclusive structure function data is sizeable, in particular at lower $x$. For these quark distributions, the additional constraints from the heavy quark data is less pronounced, though far from negligible. For the strangeness, the inclusive data already place some constraints, which are significantly improved upon by the addition of the charged-current charm data.

It is interesting, in particular from the point of view of comparing with existing LHeC PDF projections, to investigate the robustness of the results shown in Fig. 4.1 with respect to the choice of tolerance $T$ used in the analysis. As discussed in Sect. 3, here we take as a baseline a tolerance value of $T = 3$, to be roughly consistent with the underlying assumptions of the PDF4LHC15 baseline set. However, for fits to DIS–only data, as in the LHeC pseudo–data and the existing HERAPDF2.0 sets [12], a tolerance of $T = 1$ is often taken, it being argued that in this case the underlying data are cleaner and self–consistent, allowing for this textbook value to be taken. On the other hand, it has been known for some time, see e.g. [47], that fits to the HERA dataset only are found not to be consistent within the quoted uncertainties in comparison to those including collider data when using a textbook tolerance $T = 1$. Indeed, the HERAPDF2.0 result for example for the up quark at high $x$ is found to be in clear tension with global fit results [12], again indicating a tension between the underlying datasets, or more precisely the data/theory comparisons. Moreover, some degree of tension within the HERA dataset itself has also been observed, with the $e^- p$ CC data pulling in a different direction to other HERA measurements [48], and the final combined charm and beauty structure function data [49] being in some tension with the inclusive data.

Nonetheless, it is instructive for the purposes of this exercise to examine the further reduction in uncertainties when a tolerance of $T = 1$ is assumed. In Fig. 4.2 we show a similar comparison to that of Fig. 4.1 for the complete LHeC dataset, but with the baseline results obtained with a tolerance $T = 1$, together with those based on $T = 3$ instead. We can see that the difference is as expected largest where the LHeC data have more of an impact, generally leading to a further factor of around 2 reduction in the uncertainty. On the other hand at higher $x$, where the impact of the LHeC data is smaller, the difference is consequently reduced.

Note however that this comparison has been carried out for illustration purposes alone, as using $T = 1$ is actually inconsistent with the assumptions upon which the Hessian PDF sets in PDF4LHC15 are based. Indeed, the PDF uncertainties in PDF4LHC15 set correspond to an approximate value of the tolerance of $T \simeq 3$, and certainly something larger than the textbook $T = 1$ case. This implies that taking $T = 1$ for the LHeC pseudo–data in effect corresponds to weighting such data preferentially with respect to the existing data that leads to the PDF4LHC15 baseline set. This therefore does not accurately reproduce the situation of a global PDF analysis.

A further point worth clarifying is whether the reduction in uncertainties when including data from the LHeC will be significantly less when compared to a more recent global PDF fit than PDF4LHC. To this end, in Fig. 4.3 we compare the impact of the LHeC pseudodata on the PDF4LHC baseline to the result found by taking the newer NNPDF3.1 [16] set, which in particular includes a range of more recent LHC data, from high precision $W$, $Z$ to differential top quark pair and the $Z$ boson transverse momentum distribution. To be precise, we take the NNLO symmetric Hessian version of this set, to allow us to perform the same profiling exercise as before. We can see that the baseline uncertainties are indeed smaller in comparison to the PDF4LHC case, though it should be emphasised that as the PDF4LHC combination will normally have larger errors than any of the three global inputs, this would also in general be true for the earlier NNPDF set entering into the combination. Even so, the results are qualitatively rather similar to the PDF4LHC case, and the impact of the LHeC data remains clear.

We now consider how the PDF impact of the LHeC pseudo–data compares to that of the corresponding HL–LHC projections reported in [9]. Moreover, we would also like to quantify

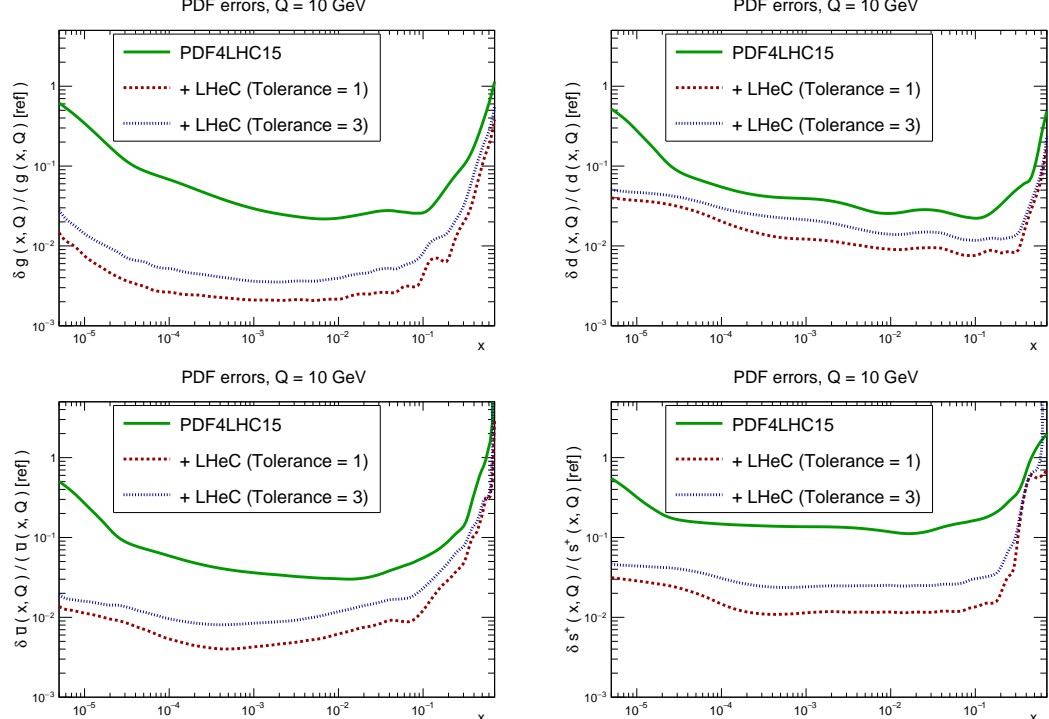

Figure 4.2: Same as Fig. 4.1 for the complete LHeC dataset, now comparing the baseline results obtained with a tolerance $T = 3$ with those based on $T = 1$.

the expected PDF uncertainty reduction if both the HL–LHC and LHeC pseudo–data are simultaneously added to PDF4LHC15 by means of profiling. In Figs. 4.4 and 4.6 we show a similar comparison to that of Fig. 4.1, including the LHeC results in addition to the HL–LHC projections as well as their combination. We can see that at low $x$ the LHeC data place in general by far the strongest constraint, in particular for the gluon. This is to be expected: the LHeC provides an outstanding coverage of the small-$x$ kinematical region, as illustrated in Fig. 2.1.

In the intermediate $x$ region, the HL–LHC and LHeC pseudo–data are found to place comparable constraints on the PDFs. At higher $x$ the constraints are again comparable in size, with the HL–LHC resulting in a somewhat larger reduction in the gluon and strangeness uncertainty, while the LHeC has a somewhat larger impact for the down and anti-up quark distributions. To show this more clearly, in Fig. 4.5 we show the same plot as before for the gluon PDF, but with a linear $x$ scale. The combination of both HL–LHC and LHeC pseudo–data nicely illustrate a clear and significant reduction in PDF uncertainties over a very wide range of $x$, improving upon the constraints from the individual datasets in a non–negligible way.

In order to further assess the PDF impact of the LHeC pseudo–data (alone or in combination with the LHeC measurements), and in particular their relevance for LHC phenomenology, in Fig. 4.7 we show the impact on the gluon–gluon, quark–gluon, quark–antiquark and quark–quark partonic luminosities. In this comparison, we display the relative reduction of the PDF uncertainty in the luminosities compared to the baseline. For example, a value of 0.2 in this plot indicates that the profiled PDF uncertainties are reduced down to 20% of the original ones.

Some clear trends are evident from this comparison, consistent with the results from the individual PDFs shown in Figs. 4.4 and 4.6. We can in particular observe that at low mass the LHeC places the dominant constraint, while at intermediate masses the LHeC and HL–LHC constraints are comparable in size, and at high mass the stronger constraint on the gluon–

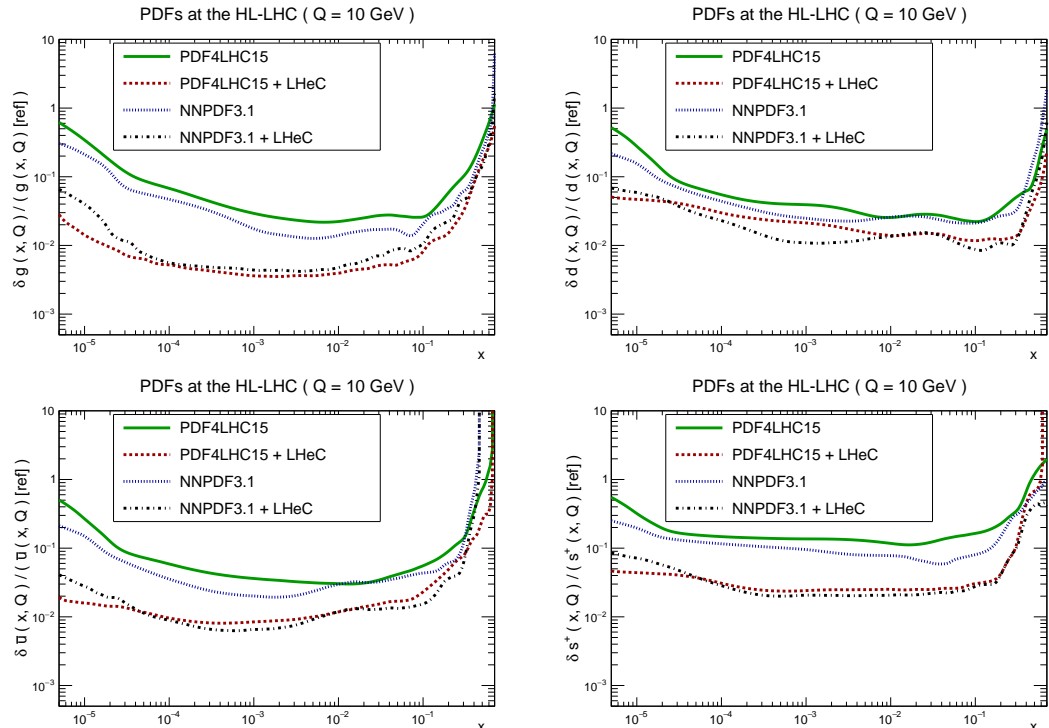

Figure 4.3: Same as Fig. 4.1 for the complete LHeC dataset, with the result of profiling the NNPDF3.1 Hessian set also shown.

gluon and quark–gluon luminosities comes from the HL–LHC, with the LHeC dominating for the quark–quark and quark–antiquark luminosities.

As in the case of the PDFs, for the partonic luminosities the combination of the HL–LHC and LHeC constraints leads to a clear reduction in the PDF uncertainties in comparison to the individual cases, by up to an order of magnitude over a wide range of invariant masses, $M_X$, of the produced final state. It is also worth emphasising that the LHeC and HL–LHC will have completely different experimental and theoretical systematics, thus their complementarity would provide a particularly precious asset to disentangle possible beyond the Standard Model (BSM) effects.

In summary, the LHeC and HL–LHC datasets both place significant constraints on the PDFs, with some differences depending on the kinematic region or the specific flavour combination being considered. Most importantly, we find that these are rather complementary: while the LHeC places the most significant constraint at low to intermediate $x$ in general (though in the latter case the HL–LHC impact is often comparable in size), at high $x$ the HL–LHC places the dominant constraint on the gluon and strangeness, while the LHeC dominates for the up and down quarks. Moreover, when both the LHeC and HL–LHC pseudo–data are simultaneously included in the fit, all PDF flavours can be constrained across a wide range of $x$, providing a strong motivation to exploit the input for PDF fits from both experiments, and therefore for realising the LHeC itself.

Finally, a few important caveats concerning this exercise should be mentioned. First, the processes included for both the LHeC and HL–LHC, while broad in scope, are by no means exhaustive. Most importantly, as mentioned in Sect. 2, for the LHeC no jet production data are included, which would certainly improve the constraint on the high-$x$ gluon. In addition, the inclusion of charm production in $e^+p$ CC scattering would further constrain the strange quark. In the case of the HL–LHC, only those processes which provide an impact at high-$x$

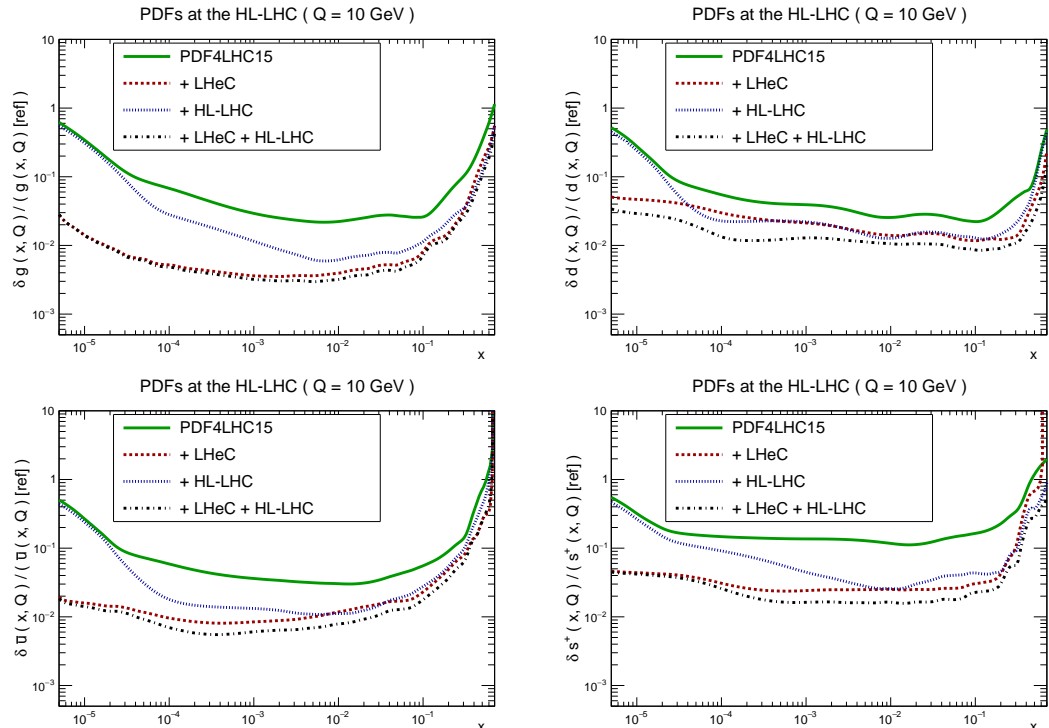

Figure 4.4: Same as Fig. 4.1, now comparing the impact of the LHeC pseudo–data with that of the HL–LHC projections and to their combination.

were included, and hence the lack of constraint at low-$x$ that is observed occurs essentially by construction. In particular, there are a number of processes that will become available with the legacy HL–LHC dataset, or indeed those in the current LHC dataset that are not currently included in global fits, but which can in principle constrain the low-$x$ PDFs, from low mass Drell–Yan to inclusive $D$ meson production [50, 51] and exclusive vector meson photoproduction [52], though here the theory is not available at the same level of precision to the LHeC case.

A further point of note is the value of the tolerance $T$ used in this analysis. By performing a closure test using PDF4LHC15 as input, we are implicitly assuming that the final LHeC (and HL–LHC) data will be describable by this set, within the $T = 3$ tolerance criteria to allow for some degree of tension among datasets. That is, one is implicitly assuming that no greater tension between datasets than this will occur. While this assumption is guided by previous experience with the wide range of measurements included in existing global PDF fits, it may turn out to be too strong and if this is the case we would expect the resulting PDF uncertainties to be larger, though at this point there is no strong motivation for believing that this will indeed be the case. On the other hand, as shown in Fig. 4.2, a more aggressive, smaller, choice for the LHeC leads to smaller uncertainties in that case, though the interpretation of such a choice in the context of this global fit to both HL–LHC and LHeC pseudo–data is far from clear, being inconsistent with the assumptions used to construct the PDF4LHC15 baseline to begin with.

# 5 Parameterisation dependence and LHeC–only fits

The main aim of this paper is to establish quantitatively the expected impact that the availability of inclusive and heavy quark structure function measurements from the LHeC would have

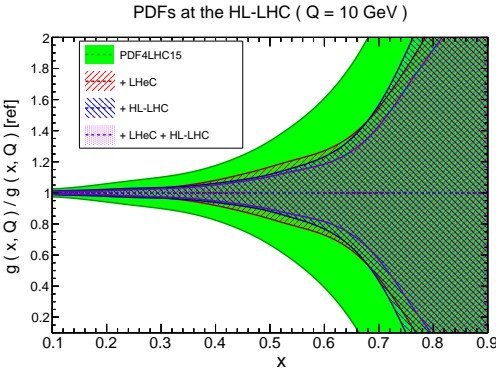

Figure 4.5: Same as Fig. 4.4, for the gluon PDF alone, with a linear scale.

on a state-of-the-art global PDF analysis. Moreover, we also want to assess how the constraints provided by the LHeC compare with those that will eventually be obtained from measurements in proton-proton collisions at the HL–LHC. To achieve these goals, we have used as a baseline the PDFLHC15 set, which contains $N_{ev} = 100$ symmetric Hessian eigenvectors, as constructed via a MC combination [47,53] of the CT14 [14], MMHT14 [13] and NNPDF3.0 [54] sets. This PDF4LHC15 set can therefore be interpreted as arising from an underlying Hessian PDF set with $N_{ev}$ free parameters determined from the experimental data.

In the current exercise we are performing a closure test, where the pseudodata are by construction in agreement with the theory generated using this PDF4LHC set. We are therefore by construction assuming that no additional parametric freedom will need to be introduced in order to describe the (future) data under consideration, or more precisely, to describe the combination of global PDF fits to these data. Such an assumption may turn out to be too strong, though as with the choice of tolerance above, there is currently no strong motivation for believing this will be true. Nonetheless, a natural question to ask is the extent to which the type of projection studies we consider here are dependent on the flexibility of the parameterisation adopted in the baseline prior PDF set.

To explore this point further, we will consider the use of a baseline PDF prior set based on a rather more restrictive parametrisation in comparison to PDF4LHC15, specifically the HERAPDF2.0 NNLO set [12]. In this HERAPDF case, there are only $\sim 14$ free parameters, reflecting the lack of constraints coming from the HERA data alone on for example the detailed quark flavour decomposition. To illustrate how this parametrization is less flexible than the one used in global fits, we note for example that the down quark valence and antiquark are parameterised in terms of only 3 free parameters, while the total strangeness is assumed to be proportional to the antidown quark. This is in contrast to the CT and MMHT sets, which have each between 2 and 3 times more free parameters in total, while the NNPDF parametric freedom is greater still.

There is therefore significantly less parametric freedom in the HERAPDF2.0 case in comparison to the PDF4LHC15 baseline. We note in particular that in the the original LHeC studies of PDF impact [10, 28] a close variant of the HERAPDF set is adopted, in terms of PDF parameterisation and quark flavour assumptions, and then a full fit is performed to the LHeC pseudo–data. In more recent studies, however, some additional freedom has been included, for example no longer assuming that the anti–up and anti–down are equal to each other at small $x$ [20, 21].

Taking into account these considerations, it follows that when the LHeC pseudo–data are generated with this more restrictive HERAPDF2.0 parametrization and flavour assumptions, one is by construction making a much stronger assumption that the future, very precise and

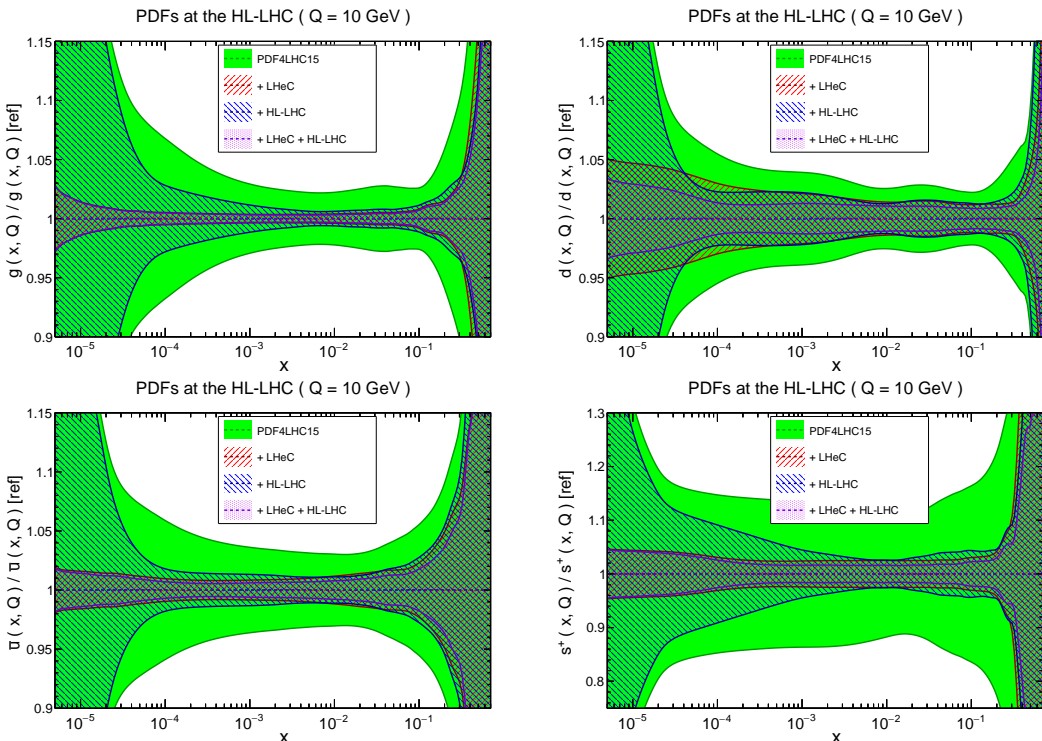

Figure 4.6: Same as Fig. 4.4, but with the error relative to each set shown.

wide ranging, LHeC data will be describable by such a restrictive parameterisation. Under this assumption, it may be expected that the corresponding projected PDF uncertainties will be smaller than in our study. To quantify this possibility, we have performed precisely the same profiling exercise as before to the same LHeC pseudo–data, but in this case using the HERAPDF2.0 NNLO set as the baseline rather than PDF4LHC15. To be consistent with the HERAPDF methodology, we take a tolerance of $T = 1$, and results are compared to the profiling of PDF4LHC15 when $T = 1$ is also used (see Fig 4.2), for a direct comparison.

In Figs. 5.1 and 5.2 we display a similar comparison to that of Fig. 4.2, now showing the impact of the LHeC pseudo–data when profiling either the PDF4LHC15 or the HERAPDF2.0 sets. Here we consider only the so-called 'experimental' component of the PDF uncertainties for the HERAPDF2.0 set, that is, the one associated with the $N_{\rm ev} = 14$ eigenvectors evaluated using the standard $\Delta\chi^2 = 1$ criterion. From this comparison, a clear systematic trend is observed, with the resultant PDF uncertainties corresponding to the profiled HERAPDF2.0 set lying significantly below the profiled PDF4LHC15 case. This effect is most pronounced for the down quark and strangeness distributions, which as discussed above are precisely those where the parametric freedom in the HERAPDF2.0 case is the most restrictive.

For the gluon PDF, which in the HERAPDF2.0 case has the largest freedom, with 5 free parameters, from Figs. 5.1 and 5.2 we observe that the differences are smaller though still rather significant. In particular, at higher $x$, one finds that the PDF uncertainty in the profiled HERAPDF2.0 case can be up to a factor 10 or more smaller than when PDF4LHC15 is adopted as the prior set. Although the uncertainty on the HERAPDF2.0 baseline is generally smaller to begin with than for PDF4LHC15, and one might therefore attribute some of the differences in the profiled uncertainties in the former case to this fact, it is worth noting that at higher $x$, where the baseline uncertainties are comparable in size, the difference is still present and indeed largest. This result thus provides a clear indication that the use of such a restricted parameterisation as in the one adopted in the HERAPDF-based prior will in general lead to

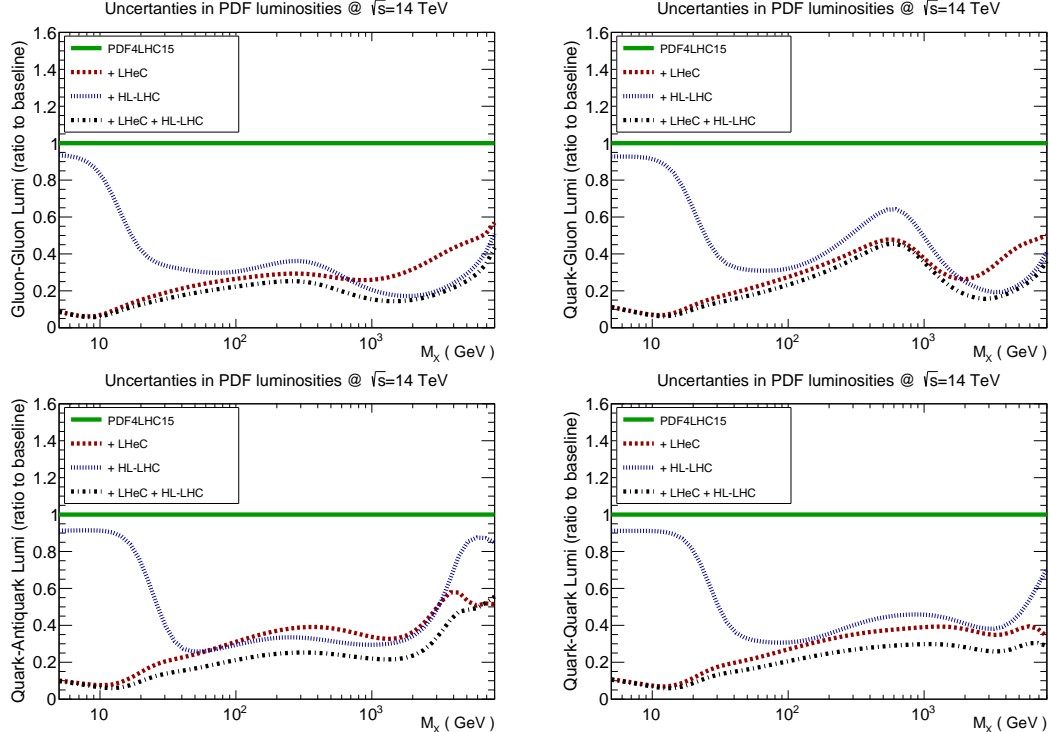

Figure 4.7: Impact of LHeC, HL–LHC and combined LHeC + HL–LHC pseudo–data on the uncertainties of the gluon–gluon, quark–gluon, quark–antiquark and quark–quark luminosities, with respect to the PDF4LHC15 baseline set. In this comparison we display the relative reduction of the PDF uncertainty in the luminosities compared to the baseline.

smaller uncertainties in the final analysis in comparison to the global fit case.

To investigate this effect further, in Figs. 5.3 and 5.4 we show the same comparison to that of Figs. 5.1 and 5.2, but in this case also including the 'model' and 'parameterisation' components of the PDF uncertainties for the HERAPDF2.0 set. Specifically, we have slightly modified the underlying PDF set to include these uncertainties in a symmetric Hessian format, rather than in the native format that requires an envelope prescription to combine them with the experimental component. These model and parameterisation components account for the effects of some additional parametric freedom, as well as for the variation of certain model parameters such as the heavy quark masses, the starting scale $Q_0$ and so on.

From this comparison, we can observe that once the additional components are accounted for, the total PDF uncertainties for the baseline HERAPDF2.0 set are now larger, and more in line with the PDF4LHC15 case. We then find that the profiled uncertainties are indeed larger than in the previous case, reflecting the additional freedom in the baseline, and lying closer to the PDF4LHC15 result. Nonetheless, there is still a clear trend for these uncertainties to lie systematically below the PDF4LHC15-based projections, reflecting the fact that even after including these additional uncertainties, the underlying parametric freedom is significantly less in the HERAPDF2.0 case. The only exception is in the low $x$ gluon (and at very low $x$ in the anti-up), where curiously the uncertainty in the HERAPDF2.0 case is in fact somewhat larger. We can find no convincing explanation of this fact, so simply leave it as an observation.

Now, in the above comparison the varying freedom in the parameterisation is not the only difference between the two baseline sets, which in addition fit to rather different underlying datasets. To clarify this, we also consider the result of reducing the relative impact of the PDF

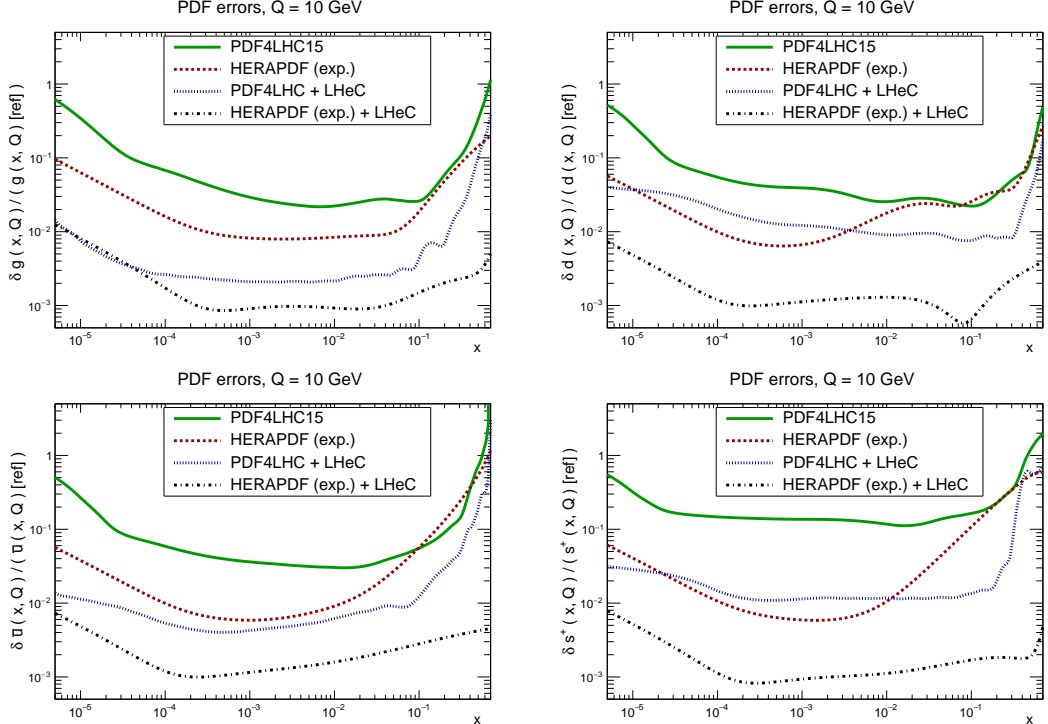

Figure 5.1: Same as Fig. 4.2, now comparing the impact of the LHeC pseudo–data when added on top of either the PDF4LHC15 or the HERAPDF2.0 sets. In both cases, a tolerance of $T = 1$ is used. For HERAPDF2.0, only experimental uncertainties are taken into account, while the model and parametric ones are not considered.

prior in the profiling, that is, the second term in (3.4), by a significant factor of 100, for both HERAPDF and PDF4LHC baselines. This should effectively remove the impact of the data that are used as input to construct the baseline sets, and their underlying PDF uncertainties. In other words, the *only* difference between these results should be in the underlying parameterisation flexibility that one assumes will be sufficient to describe the LHeC data. Moreover, as the effect of the prior datasets have been removed, in such a comparison the constraints will now be purely due to the LHeC dataset and hence in the PDF4LHC case no longer correspond to a global PDF analysis, but rather to a pure LHeC–only one, using the PDF4LHC parameterisation. We recall in particular that these comparisons are performed with a tolerance $T = 1$, and hence are directly comparable to existing LHeC–only studies [10, 17–19].

The results are shown in Figs. 5.5 and 5.6, including the PDF4LHC case with the original prior to assess the impact relative to the current PDF determination from global fits. For the PDF4LHC parameterisation, the results of the LHeC–only fit are in general significant, resulting in uncertainties that are up to an order of magnitude smaller. The one exception is the down quark, where the uncertainties are comparable; this is consistent with the fact that the determination from DIS data alone is known to be less significant in this case. Comparing with the HERAPDF case, we can again see that in general the projected uncertainties resulting from the HERAPDF baseline is significantly smaller in comparison to the PDF4LHC case. This is most marked for the 'experimental' HERAPDF baseline, while for the set including model and parameterisation uncertainties the PDF uncertainties are seen to be somewhat larger, in particular in the regions less constrained by data. A closer comparison shows that the difference between the PDF4LHC and HERAPDF errors is somewhat larger in this case, then when the baseline prior is included. This is perhaps not surprising: as the PDF4LHC15 baseline set

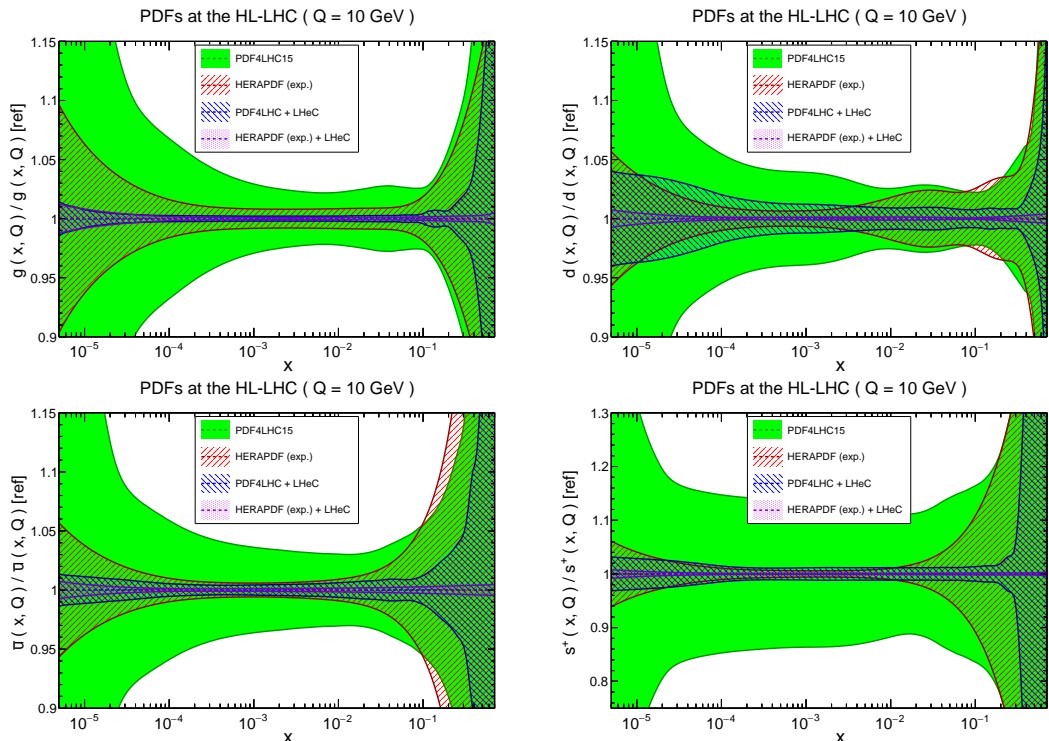

Figure 5.2: Same as Fig. 5.1, but with the error relative to each set shown.

fits to a larger dataset, the impact of reducing the prior in this case should be to increase the PDF uncertainties by more than in the HERAPDF2.0 case, leading to a larger difference.

To summarise our findings, we have demonstrated in this section how in general the quantitative interpretation of the PDF projections based on LHeC pseudo–data depends sensitively on the choice of input dataset, parametric flexibility, and flavour assumptions that enter the construction of the prior PDF fit. A more flexible parametrization, such as the one used in global PDF fits, as required to give a satisfactory description of all available measurements from lepton-hadron and hadron-hadron collisions, will in general result in larger profiled uncertainties than in the case of a HERAPDF2.0 or similar baseline. However, the latter results adopt the implicit strong assumption that the parametric flexibility of HERAPDF2.0 will be sufficient to describe all future precision measurements, including those from the LHeC. We note that such an assumption is found to be unjustified already when one accounts for the existing LHC data.

## 6 Summary and outlook

In this study, we have quantified the expected information that the realisation of the Large Hadron electron Collider (LHeC) at CERN would provide on our knowledge of the quark and gluon structure of the proton. This study, based on the pseudodata presented in [26, 27], is the first time that the impact of the LHeC measurements has been assessed in the context of a global PDF analysis. Our results complement and extend our previous study of the PDF projections based on HL–LHC pseudo-data, and provide a compelling picture of how much better our understanding of the parton distribution functions can become through the combination of these two facilities, one already fully approved (HL–LHC) and the second under consideration (LHeC). In essence, we have assessed the 'ultimate' precision that can be expected for PDFs

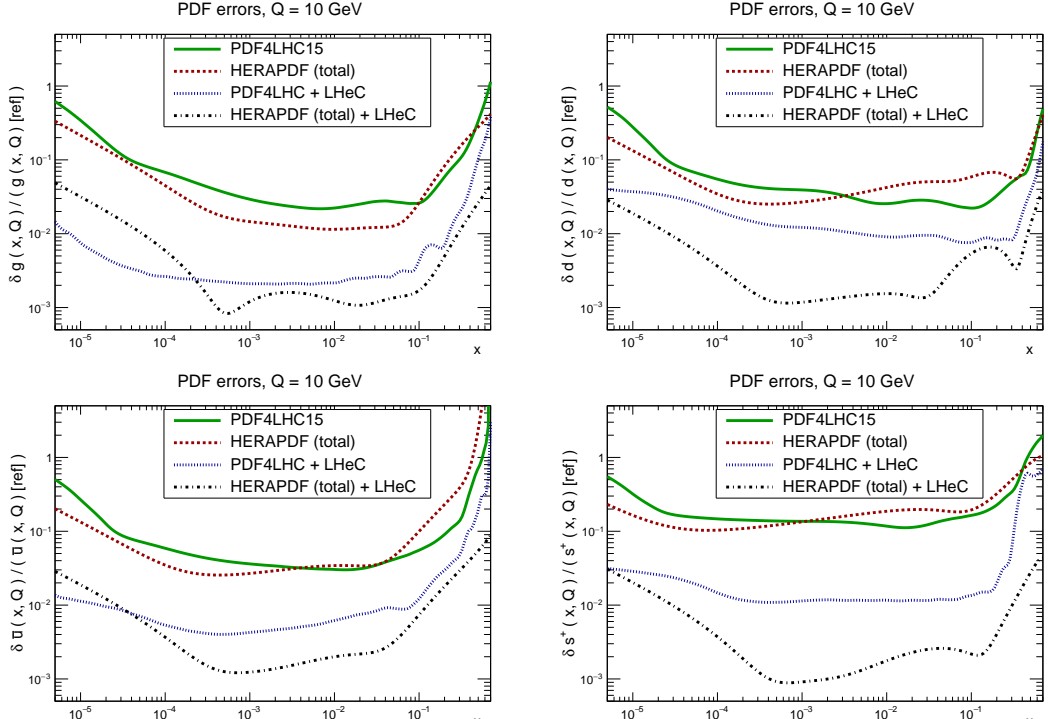

Figure 5.3: Same as Fig. 5.1, where now the baseline HERAPDF2.0 includes not only the experimental PDF uncertainty, but also the corresponding model and parametrization components.

from experiments at CERN alone by 2035. Of course, other related theory developments, such as for example progress in lattice QCD calculations [55,56], may well have a significant impact in addition.

Our results demonstrate that the LHeC can improve our current precision on PDFs significantly, with uncertainties dominated by experimental systematics, as statistical errors quickly become negligible. For those poorly constrained flavours, e.g., the gluon at both small-$x$ and large-$x$, the sea quark flavour separation, and the total strangeness, the PDF uncertainties can be expected to be reduced by up to an order of magnitude. In particular, we have shown how the availability of the strange, charm, and bottom heavy-quark production data play an important role in constraining the gluon and strange PDFs. In addition, further LHeC processes not considered here, such as jet production, could provide additional information in particular on the gluon PDF.

By comparing with the corresponding PDF projections based on HL-LHC pseudo–data, we find that the LHeC measurements would place stronger constraints in general from the small-$x$ to the intermediate-$x$ regions for most flavours. In relation to this, it is also worth emphasising that beyond studies directly relating to PDF constraints, this high precision probe of the low–$x$ region will in particular provide a unique environment to study novel QCD phenomena, such as BFKL and saturation effects. In the large-$x$ region, which is of course crucial for BSM searches, the LHeC and HL-LHC impact is broadly found to be comparable in size, with the HL–LHC resulting in a somewhat larger reduction in the gluon and strangeness uncertainty, while the LHeC has a somewhat larger impact for the down and anti-up quark distributions.

Furthermore, the combination of both the LHeC and HL-LHC pseudo–data leads to a significantly superior PDF error reduction in comparison to the two facilities individually. This could become particularly crucial for the interpretation of possible anomalies in the large $p_T$ region,

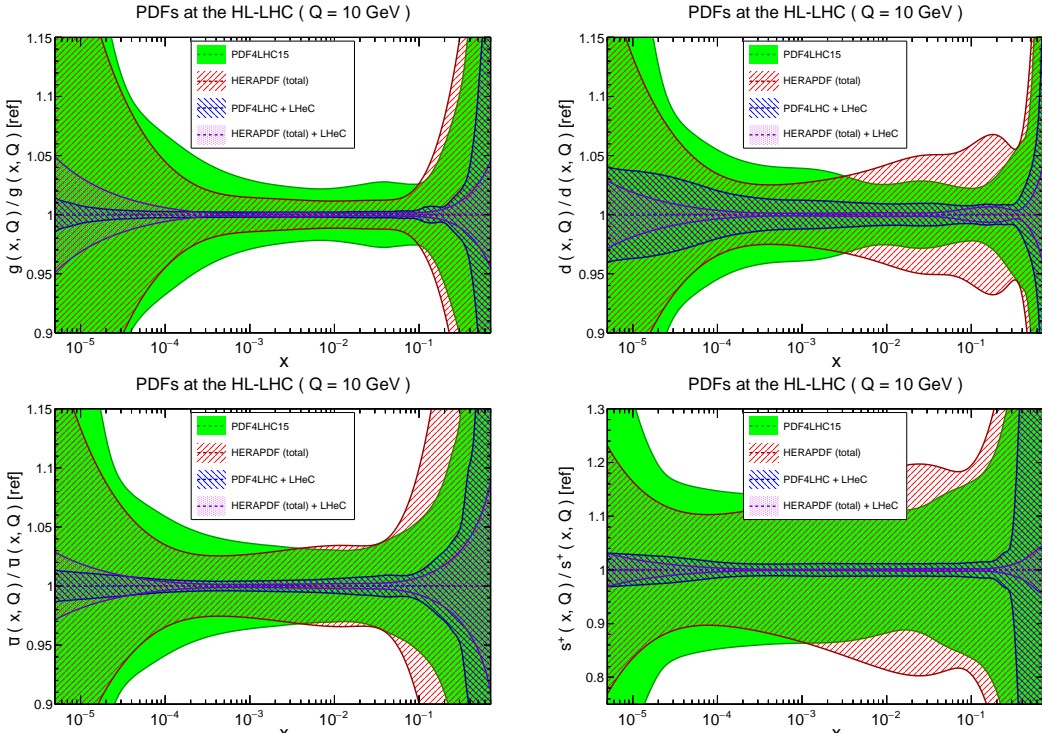

Figure 5.4: Same as Fig. 5.3, but with the error relative to each set shown.

that might indicate the presence of new physics beyond the SM. Moreover, as the complete LHeC dataset that has been used in this analysis is dominated by systematic errors, our results strongly suggest that smaller LHeC datasets, which are expected to have a similar PDF impact as the full legacy dataset, could be exploited already at relatively early stages in HL–LHC running.

In this study we have also considered the robustness of our results when using alternative PDF priors in the profiling, in particular the HERAPDF2.0 PDF set that is based on a more restrictive parameterisation in comparison to the global sets that enter the PDF4LHC15 combination. In this case, the LHeC pseudo–data in general leads to significantly smaller PDF uncertainties in comparison to the results based on the PDF4LHC15 prior. This comparison reveals the fact that, when interpreting the projections for future LHeC and HL-LHC measurements, the input PDF functional forms should be adjusted to make sure that any possible parametrization bias and flavour assumptions are minimised.

As a word of caution, in this study we have ignored any possible issues such as data incompatibilities, limitations of the theoretical calculations, or issues affecting the data correlation models. These are already common in PDF fits, but can only be addressed when carrying out a global fit with real data. The results presented in this study (as well as in any other projection) can therefore only be interpreted as providing an estimate of the possible precision on the PDFs that can be expected from these future facilities. In addition, as mentioned before, there are other possible LHeC measurements that have not been considered here, such as inclusive jet production, which can place further important constraints on the PDFs. Finally, it would be interesting to study in the future the extent to which our results might change if a full fit, rather than a profiling study, is carried out. However, it should be emphasised that within the context of the closure tests we are considering here, where data and theory agree by construction, profiling is known to provide a very good approximation to the true result.

Finally, one important point that we have not discussed in detail here is the potential for

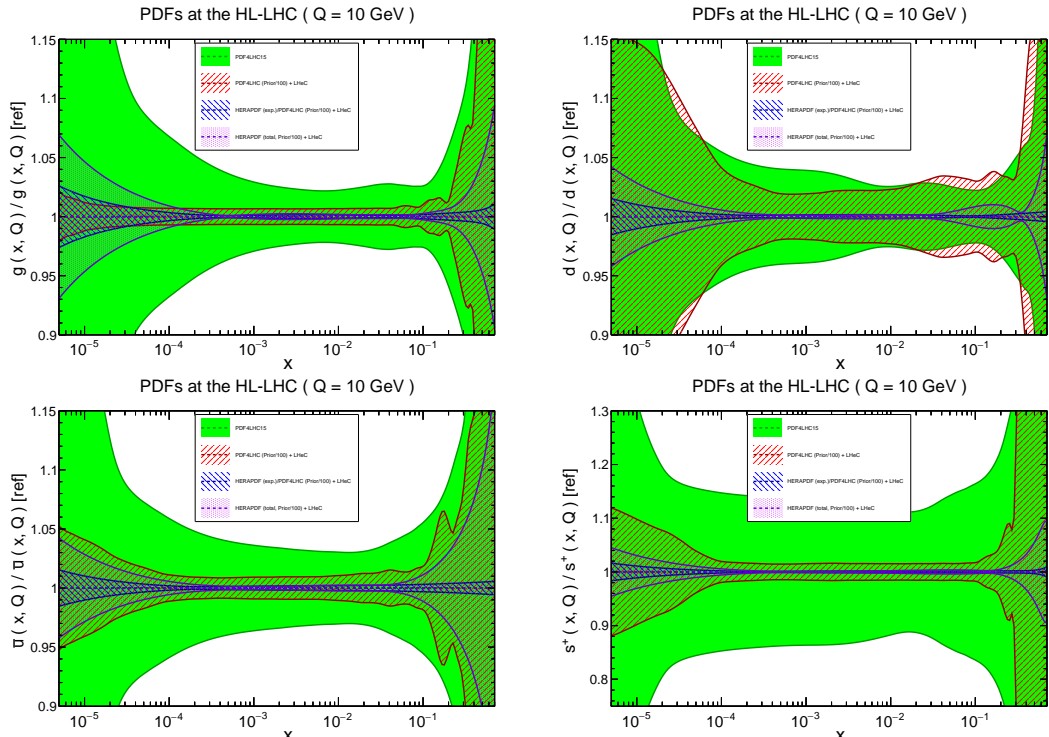

Figure 5.5: Impact of LHeC on the PDF uncertainties of the gluon, down quark, anti–up quark and strangeness distributions, with respect to the PDF4LHC15 and HERAPDF baselines, when the relative impact of the initial PDF prior, that is, of the datasets included in the baseline PDF sets, is reduced by a factor of 100. For the HERAPDF case we show results corresponding to the experimental uncertainties only baseline and that including model and parameterisation uncertainties. A tolerance of $T = 1$ is taken in all profiled cases, while the result of the PDF4LHC set (including prior but with no LHeC data) is shown for comparison.

BSM contamination of HL–LHC data at high $x$. Our projections, which are based entirely on a closure test, by construction assume that the future HL–LHC (and LHeC) data will be describable by SM theory. However, it is well known that BSM physics may enter various LHC datasets, in particular at higher $p_\perp$ and invariant masses, and hence may be incorrectly absorbed into PDF fits. A detailed study of these effects is beyond the scope of the current paper, but clearly in future fits it will be essential to disentangle BSM and QCD effects. A first recent study [29] of a fit in the framework of the SM Effective Field Theory (SMEFT) has demonstrated how this may be achieved, by using the different scaling with the energy of BSM and QCD effects as a means to separate them, complemented with other statistical estimators. Using a similar approach, it should be possible to identify potential BSM effects that might be present in the tails of LHC distributions without the risk of reabsorbing them into the PDFs. Nonetheless, this is clearly a delicate issue, and something that will be largely absent in the case of the LHeC. This provides a further strong motivation for input from such a machine in PDF fits.

The results of this study are made publicly available in the LHAPDF6 format [46] by means of Zenodo data repository:

https://zenodo.org/record/3250580

Specifically, the following PDF sets can be obtained from this repository:

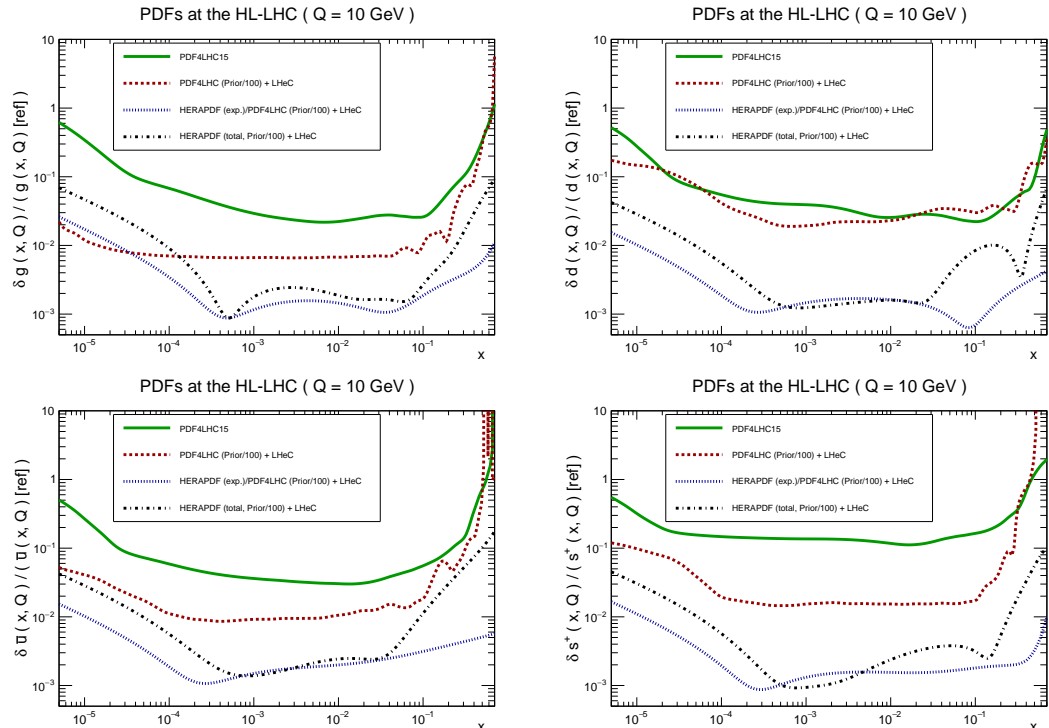

Figure 5.6: As in Fig. 5.5 but with the relative errors shown.

```
PDF4LHC15_nnlo_lhec
PDF4LHC15_nnlo_hllhc_scen1_lhec
PDF4LHC15_nnlo_hllhc_scen2_lhec
PDF4LHC15_nnlo_hllhc_scen3_lhec
```

where the first set corresponds to the profiling of PDF4LHC15 using the entire LHeC dataset listed in Table 2.1, and the other three correspond to the simultaneous profiling with both the LHeC and HL–LHC pseudo–data, for the three different projections of the experimental systematic errors. In addition, in the same repository one can also find the corresponding projections based only on the HL–LHC pseudo–data:

```
PDF4LHC15_nnlo_hllhc_scen1
PDF4LHC15_nnlo_hllhc_scen2
PDF4LHC15_nnlo_hllhc_scen3
```

In this way, the PDF projections presented in this work can be straightforwardly compared to other related projections, and used in various phenomenological applications, for example in the context of feasibility studies for future colliders both for lepton-hadron and hadron-hadron scattering.

# Acknowledgments

We thank Max Klein for providing us with the pseudo-data for the LHeC projections. We are grateful to Claire Gwenlan, Francesco Giuli, and Gavin Pownall for discussions about the LHeC pseudo–data and the xFitter-based PDF projections. We are also grateful to Nestor Armesto and Fred Olness for discussions in the context of the LHeC QCD and small-*x* working group.

**Funding information**    S. B. acknowledges financial support from the UK Science and Technology Facilities Council. L. H. L thanks the Science and Technology Facilities Council (STFC) for support via grant award ST/L000377/1. R. A. K. and J. R. are supported by the European Research Council (ERC) Starting Grant "PDF4BSM" and by the Dutch Organization for Scientific Research (NWO). The work of J. G. was sponsored by the National Natural Science Foundation of China under the Grant No. 11875189 and No.11835005.

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
