# Peer review of "Probing Proton Structure at the Large Hadron electron Collider"

_SciPost Physics, doi:SciPost Phys. 7, 051 (2019)_

## Round 1 · Referee Report · Anonymous (Referee 1) · 2019-7-23

Strengths

  1. Gives a reasonable estimate of the effect of a future LHeC on the uncertainties of Parton Distribution Functions
  2. Relates the work to other studies with critical comment
  3. Relates the work to BSM and SM cross sections of current interest
  4. Most points are made very clearly

Weaknesses

  1. Does not deal with realistic systematic uncertainties- but to be fair this is close to impossible with future projections
  2. LHeC pseudo-data are used which may be superseded soon- but that is not anything the authors could control
  3. The odd point is obscure

Report

This paper makes as assessment of the likely impact of a future LHeC on the uncertainties on parton distribution functions PDFs using similar procedures to a recent paper which assessed the impact of the HL-LHC (by an overlapping set of authors). It is a valuable contribution to the debate. It also addresses most of the criticisms that could be made of such a study within the paper itself and as such it is a pleasure to read. I particularly enjoyed the discussion of appropriate tolerances.
Even if the LHeC pseudodata used are supeseded the paper makes a contribution in comparing methods currently used to assess their significance, which will be useful in future studies.
I recommend publication and have only a few minor comments to make under requested changes, which the authors may wish to consider.

Requested changes

  1. 1 TeV proton data are mentioned in context of extending the kinematic range, would these not also be useful for the measurement of FL and hence the gluon PDF--please comment.
  2. Add to references 31 and 32 a reference to arXIV:1906.01884 on the same topic.
  3. The use of open charm and beauty production and exclusive vector meson production to constrain the low-x gluon is made. The theoretical status of these processes is not on the same level as the cross sections used and I think this should be pointed out.
  4. On page 15 it becomes a little confusing when it is stated that 'when the LHeC pseduo-data are generated with this more restrictive HERAPDF2.0 parametrisation one is making strong assumptions about the future'. Surely the LHeC pseudo-data are whatever they are independent of whether ot not PDF4LHC15 or HERAPDF2.0 are being used for the profiling exersize. The strong assumption is that the HERAPDF parametrisation will be adequate to describe future LHeC data. Can the authors please clarify, or re-phrase?
  5. In Fig 5.5 an attempt is made to separte the role of the input data set and the parametrisation. This is quite interesting but the model and parametrisation variations for the HERAPDF are not used. It would be interesting to see this figure when they are used. However since this is not the most important point of the paper, I do not insist.

  • validity: high
  • significance: high
  • originality: high
  • clarity: high
  • formatting: excellent
  • grammar: excellent

Author:  Lucian Harland-Lang  on 2019-09-05  [id 592]

(in reply to Report 1 on 2019-07-23)

Our response is found in the attachment.

Attachment:

lhec_response_ref1.pdf

---

## Round 1 · Referee Report · Anonymous (Referee 2) · 2019-7-26

Strengths

See details in extended report.

Weaknesses

See details in extended report.

Report

The paper tries to address the interesting and important question how precision deep-inelastic electron-proton data from a future 1.3 TeV collider (LHeC) could change our knowledge about the structure of the proton. This is an exercise based on projected data generated within the LHeC study group and using what the authors call a 'state- of-the art' ansatz for the determination of parton distribution functions (PDFs).

The paper presents interesting results and is rather well written. However, it has severe problems which require changes prior to a publication. These regard the relation of the authors to the origin of the pseudo-data, and they concern the distinction of using global and especially LHC data from what the LHeC provides.

Hence I like to request changes, motivated in three general points below.

1) The generated data and the section 2.1 is in fact the intellectual property of the LHeC study group with Refs. [25] and [26]. Without such an input a paper of the present type could not have been written. It is considered inappropriate that the reference to this essential input is just a webpage which is cited only once. This would be resolved and mistakes were easily resolved (such as on the positron luminosity) if, as I sincerely propose, the current authors contacted the authors of the ep pseudo- data generation and invited them to be co-authors of this paper. This would be beneficial for the paper quality, and help future collaboration. It would then justify the current wording of the section 2.1 and maintain a proper scientific standard w.r.t. intellectual property. Furthermore, details of the pseudo-data as shown in various presentations by the LHeC study group members could be added and thus the quality of the input clearly described. This would enhance the credibility of the paper considerably. A clarification of this point is of utmost importance for a publication in a peer-reviewed paper.

2) The authors claim that they are using a so-called 'state-of-the-art global fit'. This is insufficient for the scope of the paper since it misses the very important distinction of the clean theoretical basis of ep PDF sensitive data versus the plethora of global and especially pp data. The big advantage of such new, high energy and high luminosity (w.r.t. HERA) DIS data would be the self-consistent determination of all proton PDFs for the first time, and at utmost precision, including the discovery potential for new QCD phenomena such as at small x. The ground-breaking idea as presented by the LHeC study group (c.f. LHeC CDR from 2012) would be to use the independent ep data to test e.g. QCD factorisation in pp data and empower new physics searches at LHC. In contrast to this understanding, the paper in its present from presents the current (PDF4LHC) and future (HL-LHC) global PDFs as one way to determine PDFs and only asks whether LHeC would improve the precision. For theoretical, experimental and conceptional reasons, however, these are two systems which cannot just be compared without a very clear discussion of their relative merits. Briefly, the LHeC comes with the prospect to replace the global PDFs for only then one will come back to indeed test and develop QCD theory; the comparison of HL-LHC based PDFs with LHeC would then be made as an interesting check for consistency of the theory and phenomenological assumptions. Technically, and in contrast to statements in the paper, there is no need for an LHeC analysis to follow that state- of-the-art, especially it will have different parameterisations and can be expected to be internally consistent, all at N3LO, which in a global fit may be out reach for quite some time.

3) The authors refer mainly to their Ref. [9] that illustrates in a most optimistic scenario anticipated 'ultimate' PDF precision from LHC pp data. This reference is insufficient for the scope of this paper since it neglects theoretical uncertainties and current / future inconsistencies of the various pp data. Already now, we know that the current status of fitting pp data is ambiguous which is one of the main reasons why global fits require large tolerance criteria to accommodate them (known since many years already e.g. from the Tevatron jet data). For the scope of a paper which deals with such kind of new ep data, it is mandatory to formulate exactly the used parameterisations which should be adjusted to the used input sets, i.e. using also the given polarisation information would open up sensitivities to different PDFs and require thus a different set of chosen parameterisations. The before mentioned self- consistency of ep data, i.e. taken with one detector at once and analysed consistently including all correlations clearly known, demands the use of a tolerance criteria of 1. In addition to the careful description of the used parameterisations, other details like the influence of the strong coupling constant uncertainty and more details on the heavy flavour scheme are needed in the paper (including the influence of knowing masses). As also shown in the original LHeC CDR, such precise ep data could measure the strong coupling and the charm and bottom masses with very high precision (see e.g. Table 3.6 in Ref. [10] for charm mass).

Some initial, more detailed comments are listed underneath. Those should hopefully help to improve the paper further and assumes that my three general points have been addressed in a positive way.

Abstract

The abstract is missing the important difference in the theoretical grounds of ep vs pp. It also reflects the results of studying part of the expected outputs from an LHeC only, i.e. as also stated on page 20, adding ep jet data is expected to constrain the gluon distribution considerably further over a broad x range. Please modify the abstract to reflect those differences (c.f. also general comments above) and the illustrative and limited character of your study. The 'complementarity' of HL-LHC and LHeC comment is misleading since you need to compare T=1 in ep with T=3 (or any other T>1 value).

Section 1

The introduction has to reflect in more critical way the original contributions that exists already for LHeC PDF studies, e.g. Refs. [10, 26] and further updates, e.g. arXiv:1802.04317, also at workshops and conferences. Add those and other references.

It is also important to discuss critically the so-called 'ultimate' HL-LHC data and effects that new physics may have, see e.g. recent ZEUS publication about joint PDF and contact interaction fits [arXiv:1902.03048] discussing how PDFs may be modified allowing new physics contributions. This is obviously a much more critical point for pp data where new physics could be absorbed into PDFs. So, the aim to study LHeC data in addition to HL-LHC is not really relevant, as explained above, since the goal of an LHeC would be to deliver complete PDFs independently from LHC to empower pp searches.

Further on page 3 you discuss the methodology of the profiling. This is a limited methodology, however, in the context outlined above (no new physics, no new QCD dynamics), it may be used. Please add the original literature for the profiling, in particular in Chapter 3. The Ref. [9] is insufficient here and not an original source in my view.

Section 2

As stated above, the paragraph 2.1 is only valid in the present form if the colleagues who did the pseudo-data generation are appearing as co-authors. In principle the whole paragraph belongs to Refs. [25,26]! Then also mistakes appearing in the text can be corrected and the pseudo-data could be explained in detail. Correct mistakes as such: The pseudo-data do not have an eta_l cut. Table 2.1 belongs to Ref. [25]. What is F2_c,cc? The e+p data are expected to have max. luminosity of 0.3 ab-1, please respect the given inputs here. The strange contribution is generated as you can easily check http://hep.ph.liv.ac.uk/~mklein/heavyqdata/str100 Figure 2.1 needs also the reference like 'based on Ref. [25].'. The last sentence on page 6 is an inapproriate statement since the LHeC simulation framework and assumptions evolved from the 2012 CDR towards a much more flexible framework, see e.g. Ref. [19]. As the authors may know, the goal of the CDR2012 was to study the precision in the limited HERA framework, the goal of the ongoing update is to add b,c,s data and free the ubar=dbar assumption and thus demonstrate the ability to unfold the partonic contents completely. In this regard the present paper adds indeed useful and independent information.

The part 2.2 needs to be rewritten for it is essential to understand the flavour decomposition used. For experts to judge the numerical results, section 2.2 has to explain in detail all the assumptions and parametric forms for the QCD fits. The PDF4LHC ansatz appears not suitable for an LHeC pseudo-data fit. As discussed earlier, the choice of parameterisations in the PDF4LHC fit is driven by the T>1 tolerance criteria to accommodate inconsistencies in the very many used data sets. An LHeC would make such an ansatz obsolete. Furthermore, any new QCD dynamics needs to be discussed more critical, i.e. HERA data could have hints for the onset of e.g. saturation or BFKL dynamics / low-x resummation. However, an LHeC would measure such novel effects precisely and hence would give the important input how QCD theory has to be developed further. So, if the assumptions are made clearer to the reader, I think that this is a valid approach for this study to restrict themselves to test DGLAP-like inputs.

Section 3

Please cite clearly the original contributions so that the reader is guided to the direct sources (and not secondary sources).

On page 9, second paragraph, you introduce the T=3 criterion. As discussed, this is irrelevant for precision ep data. For an apple-to-apple comparison, one would need T=1 for ep and, if that is your choice, T=3 for the global fit.

Section 4

Re-phrase the use of T=1 for ep data.

Fig. 4.1 Here it has to be noted that at high x, the gluon distribution is very small and known to not better than about two orders of magnitude currently. Any delta g/g in the range of 0.5 to 1 would be a massive improvement. It would be interesting, since xg varies so dramatically, to also show xg itself, at large x in a linear x - log (xg) scale.

In the text below Fig. 4.1 you discuss 'some tensions' in HERA ep data. I am not sure why this is relevant for this study since all HERA fits use T=1 regardless of that. Then you would need to discuss the many more and much more obvious tensions in the data sets used in global fits as well and well, this would be another paper, wouldn't it. In my view, it would be much clearer and sufficient for the scope of this paper if you state the assumptions, the details of the parameterisations and then show the results.

Fig. 4.2 and discussions of T values: Please make clearer that this is more like a technical illustration only. ep data will use T=1 and in case new QCD dynamics is discovered at LHeC, new QCD theory will be used (which goes much beyond this study) .

Figs. 4.3 and 4.4 Please add the used T values in the captions. How are the relative errors are normalised? E.g. if each 'fit' gives a new central value, the fractional uncertainties may shift artificially to lower/smaller values than the reference value. Please also add that no theoretical uncertainties and other effects are considered for HL-LHC, e.g. the scale uncertainties at low and high masses are considerable.

For the discussion of LHeC vs HL-LHC: Please also repeat the assumptions: not all LHeC data fitted, no new physics in HL-LHC observed, all pp data consistent within T=3 while ep uses T=1.

The statement on page 14 on low mass Drell-Yan and inclusive D meson pp data needs a more critical discussion. Those data are prone to large theoretical uncertainties and may be finally of limited use w.r.t. improvements of our low-x knowledge.

Section 5

This section is much too general and too much focussed on current paradigms followed up in the PDF4LHC combination. It also neglects the more recent developments of the fits performed within the LHeC study group. Please rewrite it and make it specific using a rewritten section 2.2 LHeC would be a totally new machine going much beyond HERA limitations in the allowed flexible parameters. It would allow to unfold all PDFs: u,d,c,b,s (no f_s needed), ubar and dbar.

The summary of this section also needs to be rewritten.

Section 6

The sentence is unclear: "In the large-x region, which is of course crucial for BSM searches, the LHeC and HL-LHC impact is broadly found to be comparable in size, with the HL-LHC resulting in a somewhat larger reduction in the gluon and strangeness uncertainty, while the LHeC has a somewhat larger impact for the ..." -> In the case of pp, we want to find new (non-resonant like contact interaction) physics at high masses/high x, how shall we use the same data for PDF fits? This would be only possible with much more complicated fits a la the recent ZEUS paper. Please clarify what you really mean. The point to be made here is not the similar precision (which as you indicate is only due to the current neglect of jet LHeC data) but the complete change of the analysis: LHeC would deliver external, reliable precision PDF input to enable new physics searches at high mass/x, i.e. not using those data for pp PDF analyses.

Overall, the overall summary needs to be adjusted and appropriate references to the LHeC study group have to be added.

Requested changes

Please address point 1 to 3 in the extended report and further requests therein.

  • validity: good
  • significance: high
  • originality: good
  • clarity: good
  • formatting: excellent
  • grammar: excellent

Author:  Lucian Harland-Lang  on 2019-09-05  [id 591]

(in reply to Report 2 on 2019-07-26)

Our response is found in the attachment.

Attachment:

lhec_response_ref2.pdf

---

## Round 1 · Referee Report · Anonymous (Referee 3) · 2019-8-4

Report

See attachment.

Attachment

  • validity: -
  • significance: -
  • originality: -
  • clarity: -
  • formatting: -
  • grammar: -

Author:  Lucian Harland-Lang  on 2019-09-05  [id 590]

(in reply to Report 3 on 2019-08-04)

Our response is found in the attachment.

Attachment:

lhec_response_ref3.pdf

---

## Round 2 · Referee Report · Anonymous (Referee 1) · 2019-9-9

Report

I am now happy that this paper be published. I note a few 'typos' in requested changes below--basically the legends in Figs5.2,5.4 and 5.5,5.6 will not be obvious to a non-expert, or indeed if these figures are used alone, not with their companions 5.1and 5.3.
I am sure the authors will wish to correct this. I don't need to see the paper again.

I have one general comment and it concerns the authors' reply to referee 2.
The authors rightly note that T=1 may not even be applicable to HERA, with which I can- at least partially- agree, and from this deduce that it also would not be applicable to the LHeC, with which I do not agree.
The point is that at the time that HERA was running there was still a focus on finding new physics there and although measurements related to PDFs were considered very important, they were not the paramount aim. Hence insufficient attention was paid to the full consistency of methods of deriving systematic uncertainties both within and between the experiments. This is why it took until 2015 to disentangle all this information to the best of our ability- and some discrepancies remain. In the case of the LHeC the DIS cross sections needed for PDFs will/would be the paramount aim. We have learnt a lot from HERA and we would have accords as to how to handle systematics. The data will/would be analysed in a consistent manner across the whole kinematic plane, and from year to year. Hence T=1 really could be the correct tolerance.

Requested changes

Figs 5.2 and 5.4 have both lost the +LHeC from the second appearance of HERAPDF in their legends. Just compare to 5.1 and 5.3 to see what I mean.
In Figs 5.5 and 5.6 (and I thank the authors for aving added 5.6 as I requested..) I think the green line at unity is now the PDF4LHC profiling result (it was the pre-profling result in may previous figures) for comparison to the HERAPDF profling results. But the legend does not actually specify this. I think it needs to so that the figure can be used 'stand-alone'.

  • validity: -
  • significance: -
  • originality: -
  • clarity: -
  • formatting: -
  • grammar: -

Author:  Lucian Harland-Lang  on 2019-10-01  [id 613]

(in reply to Report 1 on 2019-09-09)

Please see attached response.

Attachment:

lhec_response_new1.pdf

---

## Round 2 · Referee Report · Anonymous (Referee 3) · 2019-9-12

Report

The authors have addressed all comments of the previous report in a satisfactory manner. I recommend publication.

---

## Round 2 · Referee Report · Anonymous (Referee 2) · 2019-9-19

Report

I would like to thank the authors for their replies and the implemented changes.

As the authors state, it appears that there is a basic difference in our views about what are the goals
and the potential of a new high energy electron-proton collider versus the current status for the resolution of proton structure and the development and test of QCD. The present PDF analyses essentially depend on HERA, complemented by a plethora of different, and partially incompatible LHC data. I of course fully agree with the authors that we
have to use the precious LHC data as best as we can also for constraints on the proton structure.

The LHeC, then, would have the goal to unravel not yet known secrets of the proton structure going very deep,
because of the theoretical and experimental cleanliness of DIS vs pp. The goal is to provide an ultimate
new set of PDFs with the LHeC and confront the pp data and PDF analyses of the kind the authors have been
successfully performing. This fundamentally and conceptually differs from establishing a complementarity
between ep and pp which is celebrated in the abstract and runs thru the whole paper. Like the authors
also state, there are different views, and I understand that we differ here, and acknowledge the validity of the fit results the authors present. One needs to understand that the LHC is real, HL-LHC a hopeful future and LHeC so far
only a concept one hopes will be realised. I fully acknowledge that the value of the LHeC data,
here still restricted to inclusive NC+CC and HQ simulated data, is made clear in the paper.

Please let me iterate for a moment why I am convinced about the special role of DIS vs pp. Here I like to bring forward e.g. the excellent talk by Stefano Forte in 2014 about the basic difference determine PDFs at a hadron collider versus an ep collider.
https://indico.cern.ch/event/183282/session/4/contribution/21/material/slides/0.pdf

In summary, I may also quote from the paper "On the Relation of the LHeC and the LHC"
by J. L. Abelleira Fernandez et al., arXiv:1211.5102

"In fact, because factorisation is most reliably established in deep inelastic scattering, it is the availability of precise parton distributions from lepton-hadron scattering which may allow detailed tests of the validity of factorisation for processes for which it is less well established. On the other hand, because lepton-hadron data are generally subject to smaller power-suppressed corrections, perturbatively more stable, easier to compute than most hadronic processes so results to the highest perturbative orders are available, and finally free of many complications which arise when dealing with hadronic initial and final states (such as jet definitions, or underlying event), lepton-proton data always provide a comparatively more competitive and theoretically reliable determination of parton distributions than hadron-hadron data. The natural scenario is one in which lepton-proton data are used to determine parton distributions, and the latter are then used for hadron collider processes, and there are strong reasons of principle why this is the case."

Hence I indeed request that you remove the statement about the 'complementarity of ep and pp data' e.g. remove the last sentence of the abstract and also later in the summary.

Further on the abstract it is important to realise that the high x potential of the LHeC data vs LHC has not been evaluated at the same footing because the LHeC jet data have not been available. Please add the word 'inclusive' in front of the LHeC data mentioned in the abstract (second last sentence) and elsewhere (e.g. in the summary p13) to make it very clear, in the abstract you correctly state the used data, however, the word "inclusive" will help the reader to understand clearer the origin when you present the low x and high x data results.

Minor comments are:
Table 2.1 : I see that the author of http://hep.ph.liv.ac.uk/~mklein/lhecdata/datlhecreadme
made here the comment of eta<5, however, if I check the tables of data, the actual eta of usable data is eta<4.4,
anyway, in the context of your cuts it may not matter
What I still do not understand, is if you used 1 ab-1 for e+p data or if this is a typo in the table. As Ref. [26] states, e+p luminosity is at most 0.1 ab-1. The availability of positrons in the linac-ring ep configuration is much inferior to electrons. Your results will not depend on this as you find that above a few 10 or so fb-1 the uncertainty is systematics limited.

Page 4, paragraph 2.1, line 14: In the current scenario, the charm e+p data would constrain strange like e-p would constrain anti-strange. It would be interesting to see how one may constrain s-sbar from these two measurements. One could probably use the same simulation also for e+ and consider the correlated systematics the same.

I thank for the clarification on the data simulation, while still believing that the person who provides all data would usually be a co-author of a publication. I believe this is for clarification in this case with Max Klein, and not for the referee eventually to decide.

I did not make comments to disappoint the authors but for the reason, as sketched and cited above, that with many theorists and experimentalists I hold a different opinion. In our discussion, which is known to be ongoing, it may only help if different views are expressed and explained.
To reflect and acknowledge the other view, I recommend that you add the before mentioned Ref. [arXiv:1211.5102] to your paper.

There also seems to be a misunderstanding w.r.t. used parameterisation: The parameterisation for LHeC will be determined by the LHeC data and, due to much higher statistics and range, it will surely differ, both from the HERA ansatz, also because ALL PDFs will be determined not just uv, dv, sea, c, b, xg, but as well ubar ne dbar, strange, top and all with much higher precision. It will be a "wonderful world" for QCD and PDFs and I sense there is no disagreement about that expectation with the authors.

I recommend the paper for approval with the small changes as suggested above (excluding the s,sbar fit which is a separate study) and would be grateful if the text was adjusted towards a recognition of the differences of ep and pp, which does not diminish the results currently obtained in global fit analyses such as the ones of the present authors.

Requested changes

See in Report box.

  • validity: good
  • significance: good
  • originality: good
  • clarity: good
  • formatting: excellent
  • grammar: excellent

Author:  Lucian Harland-Lang  on 2019-10-01  [id 614]

(in reply to Report 3 on 2019-09-19)

Please see attached response.

Attachment:

lhec_response_new3.pdf

---

## Round 3 · Author Response

We have modified our paper to account for the final referee's comments (see our specific responses for more information).

---

## Editorial Decision

published